# Impact of the choice of the satellite aerosol optical depth product in a sub-regional dust emission inversion

Jerónimo Escribano[1], Olivier Boucher[1], Frédéric Chevallier[2], and Nicolás Huneeus[3,4]

[1]Laboratoire de Météorologie Dynamique, Université Pierre et Marie Curie / CNRS, Paris, France
[2]Laboratoire des Sciences du Climat et de l'Environnement, CEA, Saclay, France
[3]Geophysics Department, University of Chile, Santiago, Chile
[4]Center for Climate and Resilience Research (CR)[2], Santiago, Chile

*Correspondence to:* Jerónimo Escribano (jeronimo.escribano@lmd.jussieu.fr)

**Abstract.**

Mineral dust is the major continental contributor to the global atmospheric aerosol burden with important effects on the climate system. Regionally, a large fraction of the emitted dust is produced in North Africa, however the total emission flux from there is still highly uncertain. In order to reduce these uncertainties, emission estimates through top-down approaches (i.e., usually models constrained by observations) have been successfully developed and implemented. Such studies usually rely on a single observational dataset and propagate the possible observational errors of this dataset onto the emission estimates. In this study, aerosol optical depth (AOD) products from five different satellites are assimilated one by one in a source inversion system to estimate dust emission fluxes over northern Africa and the Arabian Peninsula. We estimate mineral dust emissions for the year 2006 and discuss the impact of the assimilated dataset on the analysis. We find a relatively large dispersion in flux estimates among the five experiments, which can likely be attributed to differences in the assimilated observation datasets and their associated error statistics.

## 1 Introduction

Aerosol optical depth (AOD) retrieved from satellites is probably the most used indirect measurement of aerosol in atmospheric and climate modelling studies. The large temporal and spatial coverage of satellite AOD makes these retrievals a unique and useful product, however they cannot provide a complete four-dimensional description of the atmospheric aerosol. Data assimilation techniques have been developed to combine model and observational information in the best possible way. Their application results in new aerosol analysis and reanalysis products (e.g., Benedetti et al., 2009; Lynch et al., 2016). In the recent years, satellite-derived AOD has also been used to estimate aerosol surface emissions in the so-called *top-down* approach (e.g., Dubovik et al., 2008; Schutgens et al., 2012; Huneeus et al., 2012). This approach is often embedded in a data assimilation framework, where observations and model are systematically combined in order to estimate emissions. With these methodologies, estimates of aerosol emissions depend on the model performance, on the detail of the data assimilation system and on the quality and coverage of the observations.

Mineral dust is an important continental contributor to the global atmospheric aerosol burden (Knippertz and Todd, 2012). Airborne dust interacts with clouds, solar and terrestrial radiation and atmospheric chemistry (Atkinson et al., 2013; Mahowald et al., 2014). Deposition over the cryosphere has effects on surface albedo (Bond et al., 2013), which modulates the impact of black carbon deposition on snow and ice surfaces. Dust is a source of iron (Fe) (Jickells et al., 2005) and phosphorus (P) (Yu et al., 2015) nutrients. Therefore the deposition of dust on some continental ecosystems has impacts on the vegetation and the carbon cycle (Jickells et al., 2014). Deposition on the ocean surface can also fertilize the phytoplankton in so-called high-nutrient low-chlorophyll regions, with impacts on marine biogeochemical cycles (Wang et al., 2015). Atmospheric dust is also known to affect human health and air quality (Morman and Plumlee, 2013).

Among other uncertainties, emission fluxes of mineral dust are still highly uncertain. For instance dust emissions from the Saharan desert, a key dust region worldwide, have been estimated to range between $400 \, \mathrm{Tg \, yr^{-1}}$ (e.g., Huneeus et al., 2011) to $4500 \, \mathrm{Tg \, yr^{-1}}$ (e.g., Evan et al., 2014). While some of the uncertainty may be related to the choice of the cut-off size for dust emissions, with a larger cut-off size resulting in a larger dust emission flux and a shorter atmospheric residence time, it is nevertheless desirable to decrease the uncertainty in the dust emission flux.

Satellite observations can help reducing these emission uncertainties. The combined use of satellite observations and models may potentially lead to superior estimates of aerosol emissions (e.g., Dubovik et al., 2008; Huneeus et al., 2012). In this study we focus on the role of observations and we quantify the plausible range of emission uncertainties as a function of the chosen observational dataset. To this aim, we assimilate AOD from five different datasets in the data assimilation system presented in Escribano et al. (2016, hereafter EBCH16) with a fixed configuration for both the model and the assimilation system.

Moderate Resolution Imaging Spectroradiometer (MODIS) aerosol products have been largely used for aerosol data assimilation (e.g., Dubovik et al., 2008; Benedetti et al., 2009; Lynch et al., 2016, EBCH16). This is not surprising because the MODIS aerosol retrieval algorithms have received a lot of attention for over a decade (e.g., Remer et al., 2005, 2008; Levy et al., 2010) and, as a result, the MODIS aerosol products are of a relatively high quality (Levy et al., 2013). Over ocean and dark land surfaces, the MODIS Dark Target (MODIS-DT) algorithm is capable of retrieving AOD at visible wavelengths, while over bright surfaces AOD is retrieved through the MODIS Deep Blue (MODIS-DB) algorithm (Sayer et al., 2013). Furthermore the MODIS instrument is onboard both the Aqua and Terra satellites, with morning and afternoon overpasses, respectively, allowing for a large temporal and spatial coverage. However, MODIS products are not free of problems. Sayer et al. (2013) evaluated the latest collection of the MODIS-DB aerosol product and found a low bias in AOD over the Sahara Desert. On the contrary, it is possible that MODIS-DT is biased high over the ocean, at least in dust outflow regions (Levy et al., 2003).

Aerosol products from other satellite sensors are also suitable for use in aerosol data assimilation. In the visible spectrum, these include aerosol products from several instruments onboard low-Earth orbiting satellites like the Multiangle Imaging Spectroradiometer (Kahn et al., 2010, MISR), Polarization and Anisotropy of Reflectances for Atmospheric Sciences Coupled with Observations from a Lidar (Tanré et al., 2011, PARASOL), Advanced Along-Track Scanning Radiometer (e.g., Sogacheva et al., 2015, AATSR) and Visible/Infrared Imaging Radiometer Suite (Jackson et al., 2013, VIIRS). From geostationary satellites, AOD is available from the Spinning Enhanced Visible and Infrared Imager (Carrer et al., 2010, 2014,

SEVIRI) instrument onboard Meteosat Second Generation (MSG) and the Advanced Himawari Imager (AHI) onboard the Japanese geostationary meteorological satellite Himawari-8. In the infrared, aerosol products are available from the Advanced Infrared Radiation Sounder (Peyridieu et al., 2010, AIRS) and the Infrared Atmospheric Sounder Interferometer (Peyridieu et al., 2013, IASI) instruments, particularly for dust aerosols that have a strong signature in the longer wavelengths. Finally it is also possible to assimilate the vertical profile of the extinction coefficient from the Cloud-Aerosol Lidar with Orthogonal Polarization (CALIOP) sensor onboard the Cloud-Aerosol Lidar and Infrared Pathfinder Satellite Observation (CALIPSO) mission (Winker et al., 2009) but this is also fraught with difficulties as such inversion is fairly sensitive to assumptions made on the aerosol model.

An evaluation of some of these products is done in de Leeuw et al. (2015). The authors found that most of the compared satellite products have a good performance of AOD retrievals with respect to ground-based AOD measurements. In theory it should be possible to take advantage of their complementarity either in terms of aerosol information content or in terms of temporal and spatial coverage. In practice, assimilating several aerosol products simultaneously is fraught with difficulties because the satellite products may be inconsistent with each other, or inconsistent with the aerosol properties of the model used for data assimilation. To our knowledge there are only a few (e.g., Saide et al., 2014; Zhang et al., 2014) data assimilation studies that seek to combine different aerosol products.

In EBCH16 we described an inversion system and presented a dust source inversion for North Africa assimilating 550 nm AOD from the MODIS/Aqua instrument. We now broaden the analysis and consider several retrieval products. Rather than combining different aerosol products, we seek to understand how different aerosol products perform on their own in the data assimilation system, in order to assess the strengths and weaknesses of each aerosol dataset in the context of Saharan dust and possible inconsistencies between the products. We thus compare the assimilation of five satellite AOD retrievals with the aim to narrow uncertainties in dust emission estimates for North Africa and the Arabian Peninsula.

The next section presents the data assimilation system, the assimilated observations and the observations used in the validation. The main results and mineral dust flux estimates are shown in Sect. 3. We finish this work with our conclusions in Sect. 4.

## 2 Inversion system

Mineral dust emissions are estimated using the source inversion system described in this section. Formally, the combination of the a priori information, the AOD observations and the modelling system is done through the minimization of the following cost function:

$$J(\mathbf{x}) = \frac{1}{2}(\mathbf{x} - \mathbf{x}^b)^T \mathbf{B}^{-1}(\mathbf{x} - \mathbf{x}^b) + \frac{1}{2}(\mathbf{y} - H(\mathbf{x}))^T \mathbf{R}^{-1}(\mathbf{y} - H(\mathbf{x})) \quad , \tag{1}$$

where the variable $\mathbf{x}$ is called the control vector and is related to the aerosol emissions (Sect. 2.2); $\mathbf{x}^b$ is the prior (or background) control vector, $\mathbf{y}$ are the assimilated observations (Sect. 2.3); $H$ is the observation operator (Sect. 2.1); $\mathbf{B}$ is the covariance matrix of the background errors (Sect. 2.4); and $\mathbf{R}$ is the covariance matrix of the observation errors (Sect. 2.4).

The solution of the minimization problem is called the analysis (denoted by $\mathbf{x}^a$). In this work the *analysis AOD* is the observation operator evaluated for the analysis, that is, $H(\mathbf{x}^a)$. The components of the inversion system (the elements of Eq. (1)) and the configuration of the data assimilation system are now described.

## 2.1 Observation operator

The observation operator is described in EBCH16 and references therein. As a brief summary, the observation operator consists of the AOD estimation given by the coupling of the LMDz meteorological model (Hourdin et al., 2013) with a simplified aerosol model (Huneeus et al., 2009, hereafter referred to as SPLA). The dust emissions are calculated as in EBCH16, which itself follows the Alfaro and Gomes (2001) and Marticorena and Bergametti (1995) emission scheme. The SPLA model is an Eulerian aerosol model of intermediate complexity (Huneeus et al., 2009) with four aerosol species (fine mode aerosols, coarse

sea salt, coarse mineral dust and super-coarse mineral dust) and one tracer for gaseous aerosol precursors. In this model we parameterized the processes of boundary layer mixing, dry and wet deposition and sedimentation (for coarser particles). In the model, mineral dust aerosol is emitted in three bins. Fine mode dust has diameters less than 1 μm, coarse dust has diameters between 1 μm and 6 μm and super-coarse dust is between 6 μm and 30 μm in diameter. Once in the atmosphere, coarse and super-coarse dust are both independent model species, while fine dust is treated in the fine mode aerosol tracer. A detailed

description of the aerosol model is provided in Huneeus et al. (2009) and updated in EBCH16.

In this work, the model has been configured with 39 vertical levels, and with an horizontal zoom centered over North Africa. The horizontal resolution over North Africa is approximately 1° by 1°, and the average horizontal resolution in the zoom region (between 70°W and 70°E; and 0°N and 40°N) is approximately 1° in latitude and 1.4° in longitude. The one-year spin-up and the model simulations for the year 2006 were performed with a wind nudging based on the ERA-Interim reanalysis (Dee et al.,

2011) as explained in EBCH16.

## 2.2 Control vector

The control vector is composed of multiplicative correction factors of the model emissions as in EBCH16. These correction factors are assumed homogeneous for each element of a partition of the emission flux in space (sub-regions), time (sub-periods) and aerosol type (categories). Five categories of emissions are defined (as in EBCH16) namely i) sea salt, ii) biomass burning

emissions, iii) fine dust and coarse dust, iv) super-coarse dust, and v) fossil fuel and anthropogenic $SO_2$ emissions. In this work, correction factors of fine dust and coarse dust are lumped together, while super-coarse dust has separate correction factors. Preliminary tests have shown low sensitivity of the analysis to the grouping of the three dust correction factors in only two, either fine and coarse dust together and super-coarse independent (as in this work) or coarse and super-coarse dust lumped together and fine dust independent (as in EBCH16). Additionally, our tests show that if the three dust correction factors are

independent elements in the control vector, the assimilation results do not improve and the computational burden increases.

The same sub-regions as in EBCH16 are used; their definition depends on the emission category. For fossil fuel and anthropogenic $SO_2$ emissions and for sea salt emissions, only one global sub-region is considered. For biomass burning emissions, two sub-regions have been defined, according to a grass-like and forest-like land cover classification. For both categories of

mineral dust, 19 sub-regions have been defined: 15 over northern Africa, 3 over the Arabian Peninsula and the Middle East and one sub-region for the rest of the globe. We refer to Fig. 1 of EBCH16 for a map of the dust sub-regions.

The correction factors are assumed constant within each sub-period. Like EBCH16, sea salt has a sub-period of one year, biomass burning and fossil fuel and anthropogenic $SO_2$ emissions have a sub-period of one month. A substantial difference with EBCH16 is the length of the sub-period for dust emissions. It was set to one month in EBCH16 but is reduced in this work to three days only. With this shorter sub-period (corresponding to the sub-synoptic to synoptic scale), we expect to better capture the dust emission variability in the analysis. This results in a control vector of 4674 components (that is about 10 times larger than in EBCH16), with a $\mathbf{B}$ matrix of 4674 by 4674 elements (see Sect. 2.4). We have improved the data assimilation system presented in EBCH16 in order to deal with the larger control vector. To this effect we have carefully recoded some matrix multiplication and inversion routines, paying special attention to the computational memory management and minimizing numerical errors as much as possible. We have also applied the algorithm of Qi and Sun (2006) to ensure the semi-positiveness of some of the matrices involved in the inversion.

## 2.3 Observations

In addition to the MODIS/Aqua total 550 nm AOD retrievals that we used in EBCH16, in this study we consider a range of other aerosol products from passive instruments measuring solar reflectances. We do not consider aerosol products from passive instruments operating in the infrared or from active instruments, as they would require different observational operators, which would introduce further complications in the interpretation of the results.

We compute the analysis with the data assimilation system described in this section for five satellite retrieval datasets (MODIS/Aqua, MODIS/Terra, MISR, PARASOL and SEVIRI) for the year 2006. The assimilated observations are total AOD and fine AOD where available, that is: total AOD over ocean for all the retrievals; total AOD over land for MODIS, MISR and SEVIRI; fine AOD over ocean for MODIS, MISR and PARASOL; and fine AOD over land for MISR and PARASOL. For satellites in the "A-Train" (MODIS/Aqua and PARASOL) the sampling is done at 13:30 local time. For instruments on-board the Terra satellite (MISR, MODIS/Terra) the sampling is done at 10:30 local time. For SEVIRI, the daytime average is considered. Only observations between 70°W and 65°E in longitude and between 0°N and 40°N in latitude are assimilated.

It is necessary to note that the fine AOD derived from the satellite observations is comparable to the model fine mode AOD but there are small differences across instruments. For MODIS and PARASOL products, the fine AOD is the contribution of preselected fine mode aerosol models to the total AOD in their respective retrieval algorithms, and they are comparable (but not necessarily equivalent) to the LMDZ-SPLA fine mode AOD. For fine AOD from MISR, our post-processing of the MISR products (which is explained later) ensures the equivalence between the model and the assimilated fine mode AOD.

MODIS/Terra is a MODIS instrument on-board the low Earth orbiting satellite Terra (with equatorial overpass around 10:30 Local Time). The AOD retrievals from MODIS/Terra are calculated with the same algorithms than for MODIS/Aqua (Levy et al., 2013; Sayer et al., 2013, 2014) providing total 550 nm AOD over land (Deep Blue and Dark Target algorithms) and fine mode and total 550 nm AOD over ocean (Dark Target algorithm only). We use the Level 3 AOD merged product from Collection 6 for MODIS/Terra and MODIS/Aqua.

The POLarization and Directionality of the Earth's Reflectances instrument (POLDER, Tanré et al., 2011) onboard the PARASOL satellite measures radiances in nine narrow channels in the visible to near-infrared spectrum with up to 16 viewing geometries and information on polarization in three of the channels. Through an advanced algorithm, it reports 670 and 865 nm total AOD over ocean and 865 nm fine mode AOD over land with their corresponding Ångström coefficient. Using this coefficient, we derive the 550 nm AOD from these retrievals, for total and fine mode over ocean and fine mode over land. That is, we interpolate the AOD using the following relation:

$$\tau_{550} = \tau_{865} \left( \frac{550}{865} \right)^{-\alpha} \tag{2}$$

where $\tau_{550}$ is the AOD at 550 nm, $\tau_{865}$ the AOD at 865 nm and $\alpha$ is the Ångström coefficient between 670 and 865 nm. During year 2006, this instrument was orbiting in the "A-Train" along with the Aqua satellite. As the swath of the POLDER instrument onboard PARASOL (1600 km) is relatively close to that of MODIS (2330 km), PARASOL and MODIS/Aqua have fairly similar spatial and temporal coverage, although the two algorithms differ in the clear-sky mask they use, and hence in the spatial coverage of the AOD products.

The MISR instrument onboard the Terra satellite reports 555 nm AOD over land and ocean (Kahn et al., 2009). The MISR algorithm uses multi-angular and multi-spectral information to retrieve the AOD. The swath of this instrument is smaller than the swath of MODIS which results in less coverage. Specifically, the standard Level 2 (individual soundings) and Level 3 (daily mean maps) MISR products report 555 nm AOD for fine (less than 0.7 µm of diameter), medium (between 0.7 and 1.4 µm of diameter) and large (more than 1.4 µm of diameter) aerosols. Regrettably, the size cut-off between the MISR products and the SPLA model are not compatible, so we need to post-process the MISR products before assimilation. We do it in the following way. The MISR retrieval algorithm calculates the AOD of 74 aerosol *mixture* models in order to fit the measured radiances for each observed pixel, and the quality of the fit is estimated using a chi-square criteria (Kahn et al., 2005). Each aerosol mixture model is modelled as the weighted sum of (at most) three *basic* aerosol models. The optical properties, the two parameters of the log-normal size distribution and the relative contributions of each basic aerosol model to the mixture aerosol models are reported in the Level 2 of the MISR products along with the fitting parameters computed in the AOD retrieval. With this information and with the reported Level 2 AOD, we have calculated an estimate of the MISR 555 nm AOD with the same diameter cut-off than the SPLA model, i.e., for fine (less than 1 µm of diameter), coarse (between 1 and 6 µm of diameter) and super-coarse (larger than 6 µm of diameter) aerosols. Briefly, the post-processing of the MISR AOD consists of the following steps: (i) we calculated the contribution of each basic aerosol model to the total AOD for each observed pixel; (ii) assuming both that the reported refractive index for each model is independent of the size distribution, and that the aerosol particles are spherical, we estimated the contribution of each bin (as the SPLA definitions) to the total AOD. In this work we only used the recomputed fine mode and total 555 nm MISR AOD.

The AERUS-GEO product (Aerosol and surface albEdo Retrieval Using a directional Splitting method-application to GEO-stationary data, Carrer et al., 2010, 2014) is a full-disk daily 630 nm AOD retrieval calculated from the measured radiances of the SEVIRI instrument. These retrievals cover Europe and Africa. Unlike the above mentioned products, AERUS-GEO uses only one spectral band to calculate the daily AOD product, based on measurements done in a relatively high spatial and

temporal resolution in different (i.e., time-varying) conditions of solar angles. The native spatial resolution of this product is $3 \times 3$ km$^2$ close to the Equator. We use the 630 nm total AOD from this product. We have screened all the pixels where the "ZAge" flag of the product is greater than zero [D. Carrer, personal communication]. This filter removes suspicious large and persistent AOD values in the equatorial Atlantic Ocean which are related to a time persistency assumption in the algorithm.

After this screening, 80% and 56% of the full-disk valid data are kept over land and ocean, respectively.

    In the present work the regridding of all AOD satellite products onto the model grid was performed with a weighted-area procedure. Furthermore only the model grid-boxes covered with 30% or more of satellite valid data are considered; they are otherwise set to a missing value. This arbitrary value of 30% approximately propagates the same coverage area of the satellite products into the model grid. This regridding method successfully handles the missing values and large differences in grid

resolutions. Moreover, if the input field has no missing values and both are latitude-longitude grids, this method gives the same interpolated field as the one resulting from a bilinear interpolation.

    Figure 1 shows the average AOD for the year 2006 for each instrument described above. It is important to note the difference in the sampling time of each product. The SEVIRI product is retrieved using a combination of all the available observations per day, thus achieving a mean coverage of 75% per day in our assimilation region for the year 2006. The low Earth orbiting

satellites typically sample our region of interest only once per day. However MISR has a more narrow swath than MODIS and POLDER (on PARASOL), and so its coverage is less. The differences in the amount of successful retrievals for the instruments onboard sun-synchronous orbit satellites arise from the swath of the instruments, the amount of land retrievals, the size of the pixel associated with the details of the cloud masking algorithm that may reject more or less satellite pixels during the retrieval.

    The number of observations (after reprojection onto the model grid) assimilated here is considerably larger than those

processed in EBCH16 due to the inclusion of fine mode AOD. Specifically, the number of assimilated observations is 1,469,252 for MODIS/Aqua, 1,486,774 for MODIS/Terra, 906,949 for PARASOL, 385,235 for MISR, and 1,299,764 for SEVIRI.

## 2.4   Error covariance matrices and assimilation configuration

The covariance matrix of the background errors ($\mathbf{B}$) and the covariance matrix of the observational errors ($\mathbf{R}$) have to be prescribed in the data assimilation system. The $\mathbf{B}$ matrix is defined similarly to EBCH16; the diagonal terms of the $\mathbf{B}$ matrix

are defined using the error estimates presented in the work of Huneeus et al. (2013). These are mostly based on the range of emission estimates found in the literature, except for anthropogenic and fossil fuel emissions, which are based on the uncertainty estimates found in the literature. The standard deviation of the control vector errors (i.e., the square root of the diagonal terms of $\mathbf{B}$) is 1.3 for biomass burning emissions, 3.0 for mineral dust emissions, 2.0 for sea salt emissions and 0.18 for anthropogenic and fossil fuel emissions. We have included correlations between control vector errors. For the same

sub-region and category of dust emission (fine and coarse dust, super-coarse dust) we have defined a Gaussian correlation between sub-periods with a time-length scale of three days. In comparison with EBCH16, this shortened timescale gives more freedom to the inversion system. Along with the three day sub-periods, this timescale allows the system to have more control over the emissions, with the aim of improving the representation of individual dust events in the analysis. Furthermore, the shorter sub-period of the dust control vector of this work compared to EBCH16 (3 days versus 1 month) raises the size of the

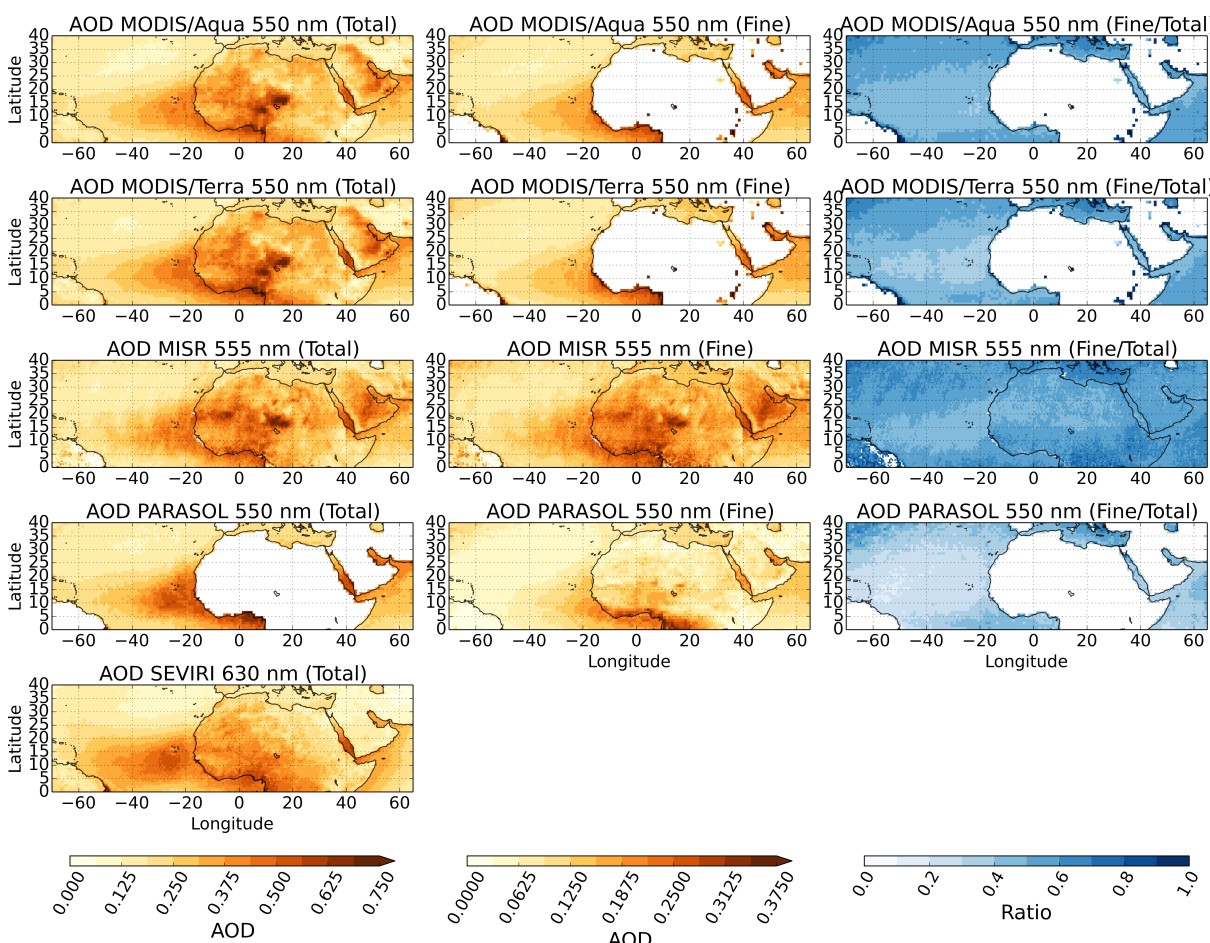

**Figure 1.** Averages for the year 2006 of the satellite-derived AOD products used in this study. The AOD products are all regridded to a regular latitude-longitude grid of 0.5° resolution for MISR and SEVIRI and 1° for MODIS and PARASOL. The total AOD is shown in the left column, the fine mode AOD (when available) in the middle column, and the ratio between the average fine mode AOD and the average total AOD is shown in the right column. Please note the 2:1 ratio of the color scales between the left (total AOD) and middle (fine model AOD) columns and the (somewhat) different wavelengths of the reported AODs.

control vector from 494 to 4,674 elements. For the same sub-region and sub-period, the correlation of errors between the fine and coarse dust emission correction factors and the super-coarse correction factor is set to 0.7.

A substantial difference to EBCH16 is the construction of the covariance matrix of the observational errors ($\mathbf{R}$). In EBCH16 the standard deviation of the observational errors was set to a fixed value of 0.2 and 0.1 for MODIS AOD products over land and ocean, respectively. In this work we keep a diagonal $\mathbf{R}$ matrix but the error statistics are defined according to the observational errors reported in the literature. A summary of these definitions is shown in Table 1. For the sake of simplicity, the errors were calculated using the satellite AOD as the reference AOD, despite the fact that most of the derivations of these error

**Table 1.** Definitions of diagonal terms in the observational error covariance matrix. The main references for the errors are shown in the table. The original error formulae were adapted for the assimilation purposes. The error shown for MODIS-DT over land is not used in this work. Errors for the SEVIRI dataset ($C_k$) are reported along with the AERUS-GEO AOD product and they are described in Carrer et al. (2010, 2014).

| Dataset | Error estimate (from reference) | Error adapted to this work | Reference |
|---------|--------------------------------|---------------------------|-----------|
| MODIS-DB | $\pm(0.03 + 0.2\tau)$ | $0.03 + 0.2\tau$ | Sayer et al. (2013) |
| MODIS-DT ocean | $[-(0.02 + 0.1\tau), +(0.04 + 0.1\tau)]$ | $0.03 + 0.1\tau$ | Levy et al. (2013) |
| MODIS-DT land | $\pm(0.05 + 0.15\tau)$ | $0.05 + 0.15\tau$ | Levy et al. (2013) |
| MISR | $\pm\max(0.05, 0.2\tau)$ | $\max(0.05, 0.2\tau)$ | Kahn et al. (2005) |
| PARASOL | $\pm 0.05 \pm 0.05\tau$ | $\sqrt{0.05^2 + (0.05\tau)^2}$ | Tanré et al. (2011) |
| SEVIRI | $\sqrt{C_k}$ | $\sqrt{C_k}$ | Carrer et al. (2010, 2014) |

formulae were done using an independent AOD dataset as a reference. For MODIS and MISR, the errors are characterized by an *expected error* (EE), which defines the boundaries of a region that contains 67% of the matchups between the satellite AOD and the reference AOD. For the MODIS merged product over land there is no equivalent error quantification. In this work, the majority of the assimilated observations over land are over North Africa and the Arabian Peninsula, where most of the
AOD is retrieved by the MODIS-DB algorithm. Hence, we adopt the MODIS-DB error quantification as the standard deviation for MODIS land AOD. Over ocean, the MODIS merged AOD is the same as the Dark Target product, but the DT EE is not centered on zero. We adopt the approximation shown in Table 1 for MODIS over ocean, shifting the EE to be symmetrical around zero at their minima. For PARASOL AOD, we assume that both terms shown in Table 1 are independent and Gaussian distributed in order to calculate the error estimate for the data assimilation system. Due to the lack of separate error estimates
of fine mode AOD, we assume the error estimates of Table 1 for fine mode AOD of MODIS, MISR and PARASOL. SEVIRI reports pixel-wise variance of the errors, which are themselves the diagonal elements of the covariance matrix of the analysis errors in the AERUS-GEO retrieval algorithm. As we do not have any information about the correlation of the errors of nearby pixels, we compute the regridded SEVIRI AOD error, assuming that all the SEVIRI pixels in the native grid are fully correlated within each model gridbox. In our case this assumption conserves the spatial structure of the AOD errors. This is done only for
SEVIRI AOD, as they report pixel-wise AOD error variance in their daily product.

Unlike EBCH16, we do not inflate the covariance matrices in order to fulfill the Desroziers et al. (2005) diagnostics. These diagnostics help detecting and correcting possible imbalances between the error covariance matrices in a data assimilation framework in the observational space. They assume that both the observations and the prior control vector do not have any bias. This assumption does not necessarily hold for all experiments in this work. Additionally, a common configuration for all
the inversions ensures a consistent methodological approach to compare the five data assimilation experiments.

As a consequence of the structure of the control vector, where fine and coarse dust correction factors are lumped together, the assimilated fine mode AOD partially constrains the coarse dust correction factor. In contrast, the super-coarse dust correction

factors are solely directly constrained by the total dust AOD. Finally, the nonzero covariances between errors of both dust correction factors propagate the assimilation of the fine mode AOD to the super-coarse dust correction factor.

## 3 Results

### 3.1 Differences and similarities in observations

Figure 1 shows the annual average for the year 2006 of the observations described in Sect. 2.3. Several characteristics that will impact the assimilation analysis can be identified in the yearly averages of the AOD. All panels clearly show the transatlantic dust plume and the local maximum of AOD in the southern Red Sea. However, maximum values of AOD over and downwind the Bodélé depression are hardly evident in the SEVIRI and PARASOL observations. For the total AOD, the SEVIRI plume over the Atlantic Ocean is more extended than in the other products. Maximum values of total AOD over the Atlantic Ocean are found close to the African coast except for SEVIRI. MODIS retrievals share similar yearly means for fine mode AOD and total AOD. In comparison, MISR AOD shows a local maximum of AOD close to 18°N, 5°W that is not observed in the other products, while an AOD local maximum at 12°N, 9°E is only observed in the MODIS products.

For fine mode AOD, there are notorious differences between PARASOL and MISR products, especially over the Sahara. PARASOL AODs are significantly smaller than MISR fine mode AOD over land and ocean.

To be able to roughly discriminate between the effect of the satellite coverage against the effect of the sampling time of the assimilated products, we have computed an equivalent of Fig. 1 but only for pairs of simultaneous AOD retrievals that correspond to (approximately) the same overpass time. These yearly averages are shown in Fig. 2. In this figure, the observations of two instruments onboard the Terra satellite (MISR and MODIS/Terra) were screened in order to compute the yearly average with pixels where both MISR and MODIS/Terra report valid data. A similar procedure was applied to the instruments onboard satellites of the A-train constellation, MODIS/Aqua and PARASOL. This screening allows a fair comparison between two pairs of retrievals.

For the colocated averages over the ocean, MODIS/Aqua and PARASOL show a similar spatial pattern for the total AOD, with colocated maxima of AOD over the Atlantic Ocean in the 5 to 15°N latitude band; both share a relatively large AOD over the Gulf of Guinea and the AOD gradient in the Red Sea (with larger values in the south). However, total AOD from MODIS/Aqua in Fig. 2 is slightly smaller than its PARASOL counterpart in the eastern transatlantic dust plume, while for the fine mode AOD, PARASOL shows smaller values.

For MODIS/Terra and MISR the differences mentioned in the description of Fig. 1 still hold when the observations are colocated (Fig. 2). Over the Arabian Peninsula, a spatial mismatch between MODIS products and MISR AOD can be identified in both Fig. 1 and Fig. 2.

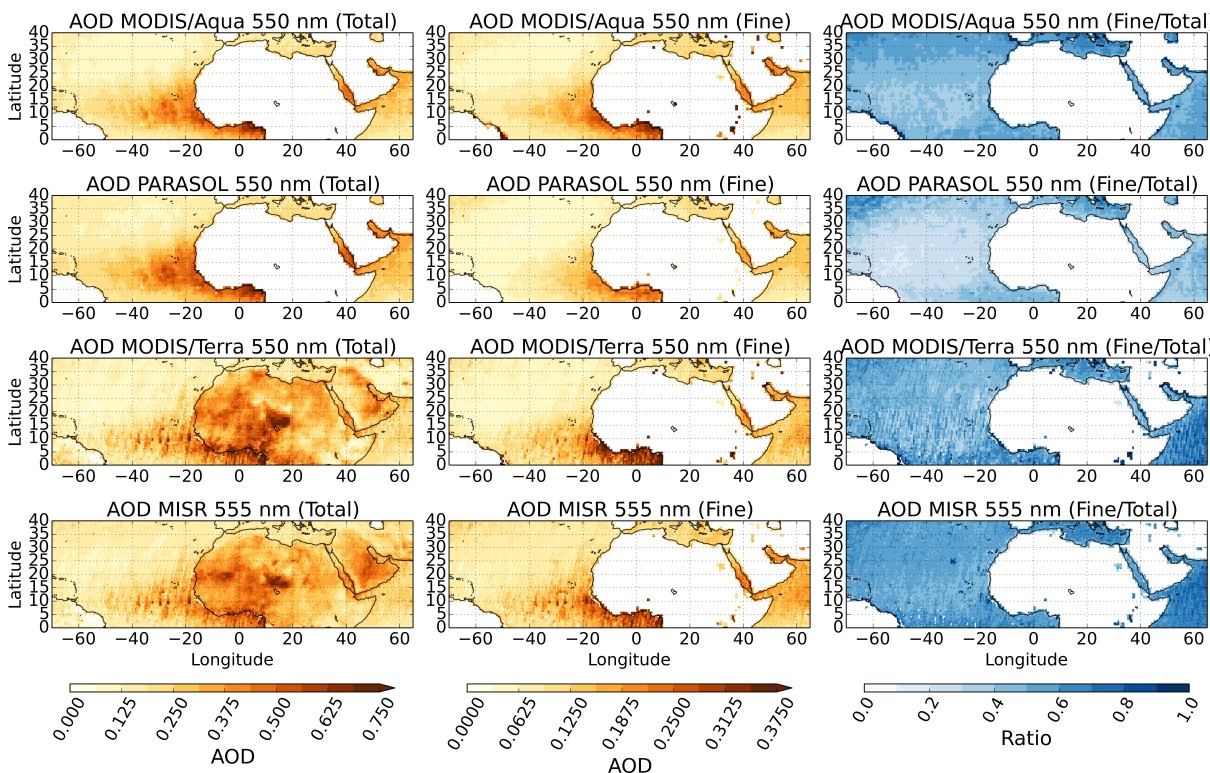

**Figure 2.** Averages for the year 2006 of the satellite-derived AOD products, similar to Fig. 1 but for colocated MISR and MODIS/Terra observations (bottom two rows), and colocated PARASOL and MODIS/Aqua observations (top two rows). The total AOD is shown in the left column, the fine mode AOD (when available) in the middle column, and the ratio between the average fine mode AOD and the average total AOD is shown in the right column. Please note the 2:1 ratio of the color scales between the left (total AOD) and middle (fine model AOD) columns.

## 3.2 Assimilation results: Departures

The assimilation performance will be explained only in terms of observation departures. Figure 3 shows histograms (in 200 bins) of the departures of the prior AOD (i.e., the difference between assimilated observations and the simulated prior AOD) and the departures of the analysis (i.e., the difference between the assimilated observations and the analysis AOD). This is shown for all five experiments. A common and expected feature of Fig. 3 is the smaller dispersion of the analysis departures with respect to the prior ones. The mode value of the histogram of the departures for the analysis is also closer to zero than for the prior in all panels (for the total AOD).

All prior histograms –except PARASOL– are slightly shifted to the right instead of being centered on zero, which means that the observations are generally larger than the prior, or in other words that the model has a low bias. This is repeated to a lesser extent in the analysis histograms for MODIS/Terra, MODIS/Aqua and MISR. For these three instruments, the land and

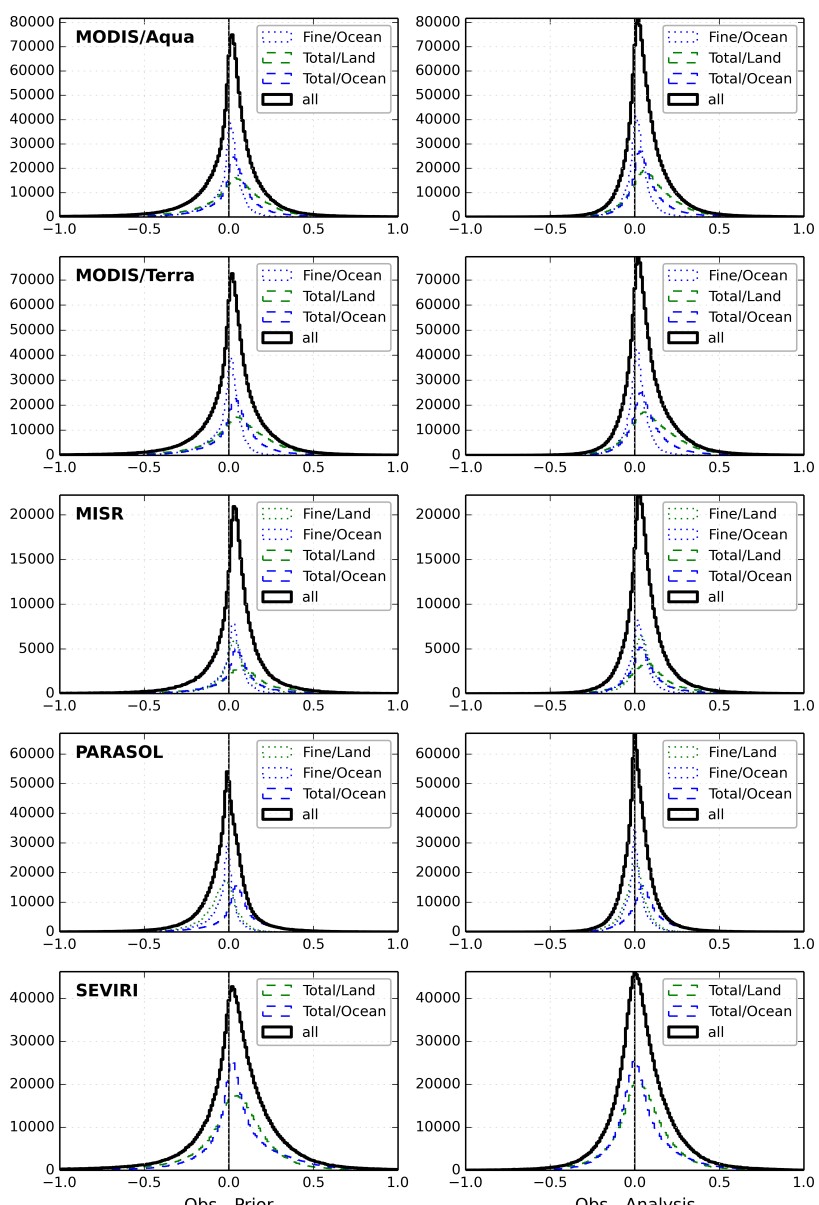

**Figure 3.** Frequency plot of departures. Observational departures with respect to the prior simulation are shown on the left column and departures with respect to the analysis are shown on the right column. Histograms are plotted between -1 and 1 in 200 bins each. Pixels over land are in green, over ocean in blue and both of them in black. Fine mode AOD in dotted lines and Total AOD in dashed lines.

ocean departures of the total AOD share similar characteristics, that is, ocean departures have less spread than land departures, and the right tails of land departures are heavier than their ocean counterparts.

The only instrument that does not have total AOD available over land is PARASOL. Departures of total AOD over ocean are larger for PARASOL than for the rest of the instruments, with a notable shift to the right, meaning that the observations are, in most of the cases, larger than the prior and analysis simulations. These large departures in the prior are mostly related to the large AOD values of the dust transatlantic plume over the eastern Atlantic Ocean.

We recall that the prior simulation is the same for all panels, and the difference in prior lies in the local time and gridboxes for which the model values are sampled. We have shown in Sect. 3.1 that, even for colocated retrievals, the geographical distribution of the AOD varies between the satellite products. We think that these differences contribute more to the differences between the histograms of Fig. 3 than the sampling differences. For example, the MODIS/Terra AOD of Fig. 1 is qualitatively similar to the MODIS/Terra AOD of Fig. 2, where only a subset of observations (which are coincident with MISR retrievals) is taken into account. On the contrary, it is easier to qualitatively observe the differences between the MISR and the MODIS/Terra panels of Fig. 2 (where both panels have the sample sampling).

A common feature is observed in all the analyses of Fig. 3, which is the preferential decrease of the left tail of the departure distributions after the assimilation. In other words, the data assimilation system is more efficient (in terms of minimizing the cost function) in decreasing larger values of model AOD than in increasing small values of model AOD. The reason for this preference is linked to the constraints imposed by the dust production model and also to the definition of the control vector. The dust production module emits dust only if some conditions are met, for example, only when there is no vegetation, the wind speed is above a threshold value (depending on the soil texture), etc. These conditions are parameterised in the model, so they depend on the model performance, but it is important to note that these conditions are based on the physical mechanisms of the natural emissions of dust. The control vector is, in practice, a multiplicative factor for the aerosol emissions. If the dust production model has no positive emission flux, the analysis cannot increase these emissions. On the contrary, if the dust emission flux is too large, the analysis can decrease the emissions. In consequence, we think that the preferential decrease of the left tail of the departure distributions is due to deficiencies of the prior in simulating some dust emissions events.

Validation against Aerosol Robotic Network (AERONET Holben et al., 1998) is qualitatively similar to the one shown in EBCH16 for all the experiments. A table summarizing the main statistics for each experiment is included in Appendix A. We would like to stress that, even though the mode of the departures is closer to zero in the analyses, the average of the departures is not necessarily closer to zero. For MODIS/Aqua, MODIS/Terra and MISR, the average of the departures for the *all* curve of Fig. 3 is larger in the analyses than in the prior. This means that for these experiments (as the average of the prior departures positive), the average AOD in the analyses is smaller than the prior AOD. This is exemplified in the comparison with AERONET, in the Appendix A, and will be related with the overall decrease of analysed emissions in Sect. 3.4.

## 3.3 Analysis AOD

Figure 4 shows the simulated 550 nm AOD for the prior and the five analyses. Larger AOD values are simulated in boreal summer (June-July-August or JJA) for all analyses and the prior. Compared to the prior, the MODIS, MISR and SEVIRI analyses decrease AOD in the northern Sahara. This is not the case for the PARASOL analysis in JJA and in boreal spring (March-April-May or MAM). There is not a large difference in AOD when the two MODIS analyses are compared between

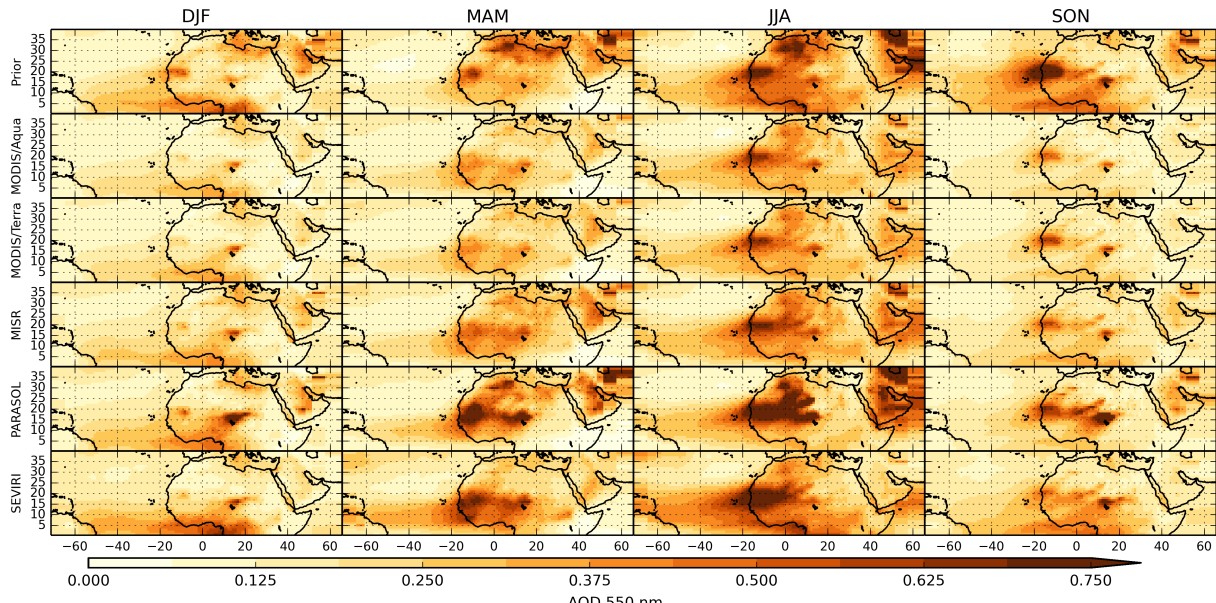

**Figure 4.** Simulated AOD at 550 nm for the prior and for the five analyses. The panels show the averaged AOD for each experiment (rows) over the months indicated in the head of the columns. MAM stands for March, April and May; JJA for June, July and August, SON for September, October and November and DJF for December, January and February. In the latter we include the first two months and the last month of the year 2006.

them, which is consistent with the discussion of the observations in Sect. 3.1. AOD from the MISR assimilation is larger in MAM than in the MODIS analysis.

In the PARASOL analysis the assimilation system increases the coarser dust emissions in order to improve the fit over the ocean. As PARASOL does not report total AOD over land, dust emissions of the coarser dust bins (and thus also with the shorter atmospheric residence times) are not fully constrained by near-source observations. This results in a large and possibly unrealistic increase in coarser mode dust emissions. For this reason we exclude this dataset from our emission flux analysis.

The SEVIRI analysis shows a larger transatlantic dust plume in MAM and JJA along with larger values of AOD over land. Observational uncertainties for SEVIRI are generally larger over land than over ocean. This allows the assimilation system to favour a better fit of the AOD over the ocean than over land. Over the transatlantic dust plume, the assimilated AOD is larger than the prior AOD. The analysis decreases this AOD difference by increasing the dust emissions in West Africa, and therefore the SEVIRI analysis shows larger AOD values over land.

## 3.4 Mineral dust flux

Mineral dust emissions were estimated with the data assimilation system using the five satellite products one by one. Total estimated flux over the Sahara and the Arabian Peninsula are shown in Table 2. Excluding the PARASOL analysis, the total mineral dust fluxes for the year 2006 ranges between 2547 and 4210 Tg. We recall that these estimates are for emitted dust

**Table 2.** Total emission flux by region and by observational dataset for the year 2006 in $\mathrm{Tg\,year^{-1}}$. AP stands for Arabian Peninsula. Western Africa refers to the longitude band between the Atlantic coast and approximately $16°$E corresponding to regions 01 to 09 in EBCH16. East Africa refers to regions 10 to 16 in EBCH16, that is, to a longitude band between approximately $16°$E and the Red Sea.

| | Prior | MODIS/Terra | MODIS/Aqua | MISR | PARASOL | SEVIRI |
|---|---|---|---|---|---|---|
| Total AP+Africa | 6657 | 3267 | 2697 | 4210 | 15748 | 2547 |
| Total Africa | 4085 | 2788 | 2361 | 3011 | 9447 | 2404 |
| Total AP | 2571 | 478 | 337 | 1198 | 6301 | 143 |
| Total Africa West | 3161 | 1808 | 1484 | 1948 | 6672 | 1544 |
| Total Africa East | 924 | 980 | 877 | 1063 | 2775 | 860 |
| Fine and Coarse AP+Africa | 1087 | 644 | 630 | 845 | 874 | 670 |
| Fine and Coarse Africa | 709 | 452 | 431 | 568 | 527 | 567 |
| Fine and Coarse AP | 378 | 192 | 199 | 277 | 347 | 103 |
| Fine and Coarse Africa West | 526 | 294 | 290 | 362 | 357 | 379 |
| Fine and Coarse Africa East | 183 | 158 | 141 | 206 | 170 | 188 |
| Super-coarse AP+Africa | 5570 | 2623 | 2067 | 3365 | 14873 | 1877 |
| Super-coarse Africa | 3376 | 2336 | 1930 | 2443 | 8920 | 1837 |
| Super-coarse AP | 2193 | 287 | 138 | 921 | 5954 | 39 |
| Super-coarse Africa West | 2635 | 1514 | 1194 | 1586 | 6314 | 1165 |
| Super-coarse Africa East | 741 | 822 | 736 | 875 | 2605 | 672 |

particles in a diameter range between 0.06 and 30 μm. The emission estimate is highly dependent on the size cut-off of the emitted particles. For airborne dust smaller than 6 μm of diameter, the total flux is estimated between 630 and 845 Tg for the year 2006. The range is therefore much smaller when we exclude the largest dust mode. Table 2 shows detailed estimates for these categories and for three geographical regions: Western North Africa, Eastern North Africa and the Arabian Peninsula.

5    Similarly to the emissions presented in Laurent et al. (2008), Western Sahara has larger emissions than Eastern Sahara. This is indeed the case in all the analyses. For both fine and coarse dust emissions, the contribution of the Arabian Peninsula is significant, indicating that is an important dust source even though it does not receive much attention in the literature. However, super-coarse dust emissions of the Arabian Peninsula are, in general, one order of magnitude smaller than North African emissions.

10    Figure 5 shows emission fluxes split by month for the three bins of SPLA. It can be seen that most of the dust emission flux is achieved in the super-coarse size range. For the reasons explained above, super-coarse dust emissions of the PARASOL analysis are much larger than expected. However, this is not the case for the coarse dust flux of the PARASOL analysis due to the structure of the control vector, where the fine and coarse dust correction factors are lumped together. As it was the case in EBCH16, the dust emission fluxes from the analysis are systematically smaller than for the prior simulation, for almost all

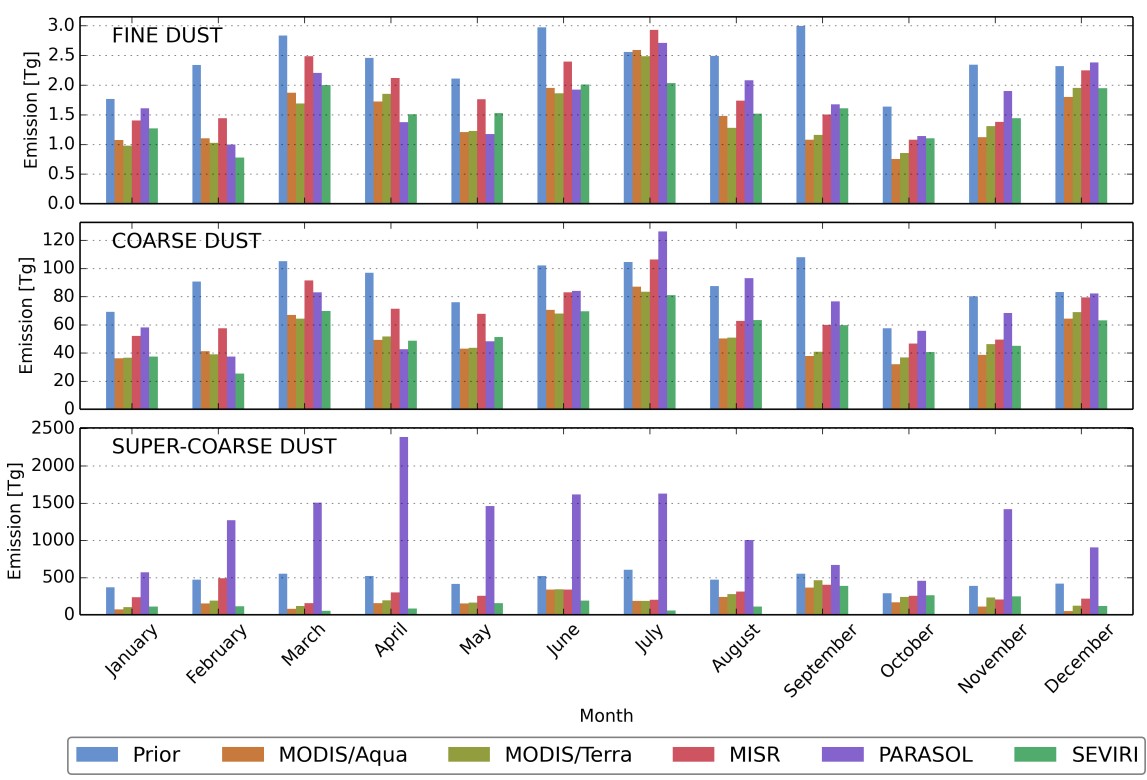

**Figure 5.** Total dust flux per month over the Sahara and the Arabian Peninsula. Fine mode dust is shown in the first panel, coarse mode dust in the middle panel and super-coarse mode dust in the lower panel. The different bars show the total mineral dust flux over the Sahara and the Arabian Peninsula by experiment and month. Note that the three plots use different scales.

dust bins, regions and months. This is largely noticeable for the super-coarse dust emission panel. The decrease of emissions of the analyses with respect to the prior is consistent with the results discussed in Sect. 3.2, where the average AOD is smaller in the analysis than in the prior, for the simulated AOD coincident with the assimilated observations for the MODIS and MISR experiments.

5    In general, coarse and fine dust emissions have maximum values in July, June, March and December while the super-coarse dust emission peaks in September. Throughout the year, coarse and fine dust fluxes share the same emission cycle, indicating consistent seasonality across the various assimilated observational datasets. However, we cannot completely rule out that a model bias (at the seasonal scale) generates this feature.

Sensitivity to the observation sampling time and coverage is not explored in this work explicitly, but the impact of the 10   sampling time can be inferred to some extent from a comparison between the two MODIS analyses. Both MODIS retrievals are expected to have similar performance when compared against reference datasets (Levy et al., 2015; Sayer et al., 2015). Our results indicate that, despite the relatively large spread (hundreds of Tg per year) in the overall analysed dust flux from the two instruments, their seasonal cycles are similar. If both instruments are unbiased (or at least if they have the same bias),

the sampling time of the products would be the most important difference in the data assimilation system. In this case, the mismatch on the overall emission flux, which is controlled by emissions from the super-coarse dust, can be likely attributed to the representation of the diurnal cycle of model emissions and boundary layer processes.

## 4   Conclusions

We have assimilated AOD from five satellite retrievals into a common data assimilation system. The control vector elements consist of correction factors for the prior aerosol emission flux over sub-regions of the Sahara and the Arabian Peninsula. Observational error statistics were adapted from the literature. For four of the five AOD datasets, fine mode AOD was assimilated when it was available. As expected, the analysis departures are, in general, smaller than the prior departures. The a posteriori estimated mineral dust flux shares a common seasonal variation between the various data assimilation configurations, but there is a relative large spread in the yearly total amount. This work estimates a total amount of emitted mineral dust over North Africa and the Arabian Peninsula ranging between 2550 and 4210 $\mathrm{Tg\,yr}^{-1}$, for mineral dust particles smaller than 30 μm of diameter in the year 2006. For mineral dust smaller than 6 μm of diameter, the estimated flux is between 630 and 845 $\mathrm{Tg\,yr}^{-1}$.

We isolated the role of the assimilated observation dataset (by freezing the rest of the inversion configuration) and showed that the large spread of these fluxes is likely associated to differences between these datasets (including their associated error statistics) rather than to model biases or deficiencies in the data assimilation system. This is despite the satellite AOD observations being of similarly good quality (or at least perceived as such). The dust emission fluxes are nevertheless sensitive to model biases or missing or under-represented processes in the model. In fact, the large emission of super-coarse dust in the PARASOL experiment could indicate that the model is not able to reproduce airborne dust transport and removal processes well. For this product, a coarse mode AOD retrieval over land would be beneficial in the assimilation.

Despite the fact that MISR has a smaller swath compared to the other assimilated products, the capability to report total and fine mode AOD over land is beneficial to the assimilation. This can be seen when the analysis was compared against AERONET AOD (Appendix A); the MISR analysis skills are similar to the rest of the analyses although the number of assimilated observations is smaller.

It is important to maintain the variety of current AOD retrieval approaches, explored by different groups with different algorithms, while improving the quality and achieving some convergence (through error reduction of the individual products). There are however two limitations in our treatment of observational errors due to the lack of information about the assimilated products. First, the assimilated fine mode AOD error variance was assumed to be similar to the total AOD error variance. Indeed, the characteristics of fine mode AOD errors are unknown, but this information would be useful and could, in principle, improve the analysis. Secondly, we assumed uncorrelated errors between fine and total assimilated AOD. As both AODs are computed simultaneously in the retrievals using similar hypotheses and radiance measurements, this assumption does not necessarily hold. Ideally, these statistics should be provided by the retrieval algorithm and reported along with the observations. Likewise it would be useful to consider error covariances in space (and possibly in time). A new generation of aerosol retrieval algorithms

based on statistically optimized fitting of observations, such as that of GRASP (Dubovik et al., 2014), can in principle provide such information. It would be interesting to test the impact of including such improved error statistics in the source inversion.

The year-to-year variability of dust emission fluxes was not considered in this study. It could increase or decrease the spread in dust emission flux estimates. Although different satellite aerosol instruments are available for different periods, there are

sufficient overlaps between instruments to gain understanding from multi-year retrievals.

Finally, reducing modelled and observational biases is another key to improve top-down emission flux estimates. Pope et al. (2016) evaluated the analysis increments in a data assimilation framework and found that large increments were associated with meteorological conditions for which the model lacks performance. Another approach which we leave for future work would be to estimate the net aerosol fluxes, that is, including variables related to the aerosol removal processes in the control vector. It

would be interesting to explore this approach, since bias in the aerosol removal processes could introduce bias in the emissions if only the emissions are optimised; but the implementation of this data assimilation could be be difficult to accomplish, due to the increase in the degrees of freedom in an ill-posed data assimilation problem.

## Appendix A: Comparison with AERONET

For validation, we select AERONET stations in the same way as in EBCH16. We only consider stations with at least 182 valid

daily 500 nm AOD retrievals of Level 2 product (Version 2). The following stations meet this criteria for the year 2006 in the region of interest: Bahrain, Blida, Dhabi, Dhadnah, Forth Crete, Granada, Hamim, Ilorin, La Parguera, Nes Ziona, Santa Cruz Tenerife, Sede Boker and Solar Village. The model AOD is recomputed at 500 nm for comparison with the AERONET AOD. The summary of statistics is shown in Table A1.

*Competing interests.* The authors declare that they have no conflict of interest.

*Acknowledgements.* The authors would like to thank the MODIS, MISR, PARASOL, AERUS-GEO and AERONET teams for making their retrieval available, F.-M. Bréon and D. Carrer for their advice with the POLDER/PARASOL and the AERUS-GEO retrievals, respectively. The POLDER/PARASOL and AERUS-GEO data were downloaded from the ICARE analysis and data centre (http://www.icare.univ-lille1.fr/). MODIS AOD products are available at http://modis-atmos.gsfc.nasa.gov, MISR AOD were downloaded from the Atmospheric Science Data Center at NASA (https://eosweb.larc.nasa.gov/) and AERONET AOD is available at http://aeronet.gsfc.nasa.gov. Input soil

data used in this study is available at http://www.lisa.univ-paris12.fr/mod/data/index.php. The work was co-funded by the project OSIRIS from MEDDE/INSU, the Copernicus Atmosphere Monitoring Service, implemented by the European Centre for Medium-Range Weather Forecasts (ECMWF) on behalf of the European Commission, and by the France-Chile ECOS project number C14U01. Part of the work was done using computing time from the TGCC under the GENCI projects t2014012201, t2015012201 and t2016012201. Nicolás Huneeus acknowledges support from FONDAP 15110009 and FONDECYT 1150873.

| Station | | Bahrain | Blida | Dhabi | Dhadnah | Forth Crete | Granada | Hamim | Ilorin | La Parguera | Nes Ziona | Santa Cruz Tenerife | Sede Boker | Solar Village |
|---|---|---|---|---|---|---|---|---|---|---|---|---|---|---|
| Latitude (°N) | | 26.21 | 36.51 | 24.48 | 25.51 | 35.33 | 37.16 | 22.97 | 8.32 | 17.97 | 31.92 | 28.47 | 30.86 | 24.91 |
| Longitude (°E) | | 50.61 | 2.88 | 54.38 | 56.32 | 25.28 | -3.6 | 54.3 | 4.34 | -67.05 | 34.79 | -16.25 | 34.78 | 46.4 |
| Elevation (m.a.s.l.) | | 25 | 230 | 15 | 81 | 20 | 680 | 209 | 350 | 12 | 40 | 52 | 480 | 764 |
| N obs. | | 201 | 195 | 243 | 324 | 283 | 276 | 263 | 270 | 251 | 185 | 233 | 335 | 335 |
| Mean | Obs. | 0.433 | 0.258 | 0.434 | 0.404 | 0.196 | 0.177 | 0.314 | 0.705 | 0.148 | 0.226 | 0.171 | 0.2 | 0.372 |
| | Prior | 0.472 | 0.313 | 0.411 | 0.454 | 0.273 | 0.209 | 0.346 | 0.434 | 0.145 | 0.245 | 0.178 | 0.263 | 0.37 |
| | MODIS/Aqua | 0.304 | 0.179 | 0.236 | 0.242 | 0.176 | 0.127 | 0.196 | 0.319 | 0.113 | 0.14 | 0.119 | 0.168 | 0.276 |
| | MODIS/Terra | 0.309 | 0.187 | 0.238 | 0.241 | 0.184 | 0.133 | 0.196 | 0.329 | 0.116 | 0.151 | 0.126 | 0.187 | 0.289 |
| | MISR | 0.423 | 0.21 | 0.316 | 0.322 | 0.212 | 0.154 | 0.255 | 0.374 | 0.129 | 0.187 | 0.142 | 0.264 | 0.376 |
| | PARASOL | 0.452 | 0.197 | 0.349 | 0.372 | 0.166 | 0.129 | 0.286 | 0.379 | 0.109 | 0.141 | 0.138 | 0.204 | 0.449 |
| | SEVIRI | 0.267 | 0.22 | 0.208 | 0.203 | 0.222 | 0.173 | 0.188 | 0.487 | 0.194 | 0.192 | 0.166 | 0.219 | 0.267 |
| Bias | Prior | 0.04 | 0.056 | -0.023 | 0.05 | 0.076 | 0.032 | 0.032 | -0.271 | -0.003 | 0.019 | 0.007 | 0.063 | -0.002 |
| | MODIS/Aqua | -0.128 | -0.079 | -0.198 | -0.162 | -0.02 | -0.05 | -0.118 | -0.386 | -0.035 | -0.085 | -0.052 | -0.031 | -0.097 |
| | MODIS/Terra | -0.123 | -0.071 | -0.196 | -0.163 | -0.012 | -0.044 | -0.118 | -0.376 | -0.032 | -0.075 | -0.045 | -0.013 | -0.083 |
| | MISR | -0.01 | -0.048 | -0.118 | -0.082 | 0.016 | -0.023 | -0.059 | -0.331 | -0.019 | -0.039 | -0.029 | 0.064 | 0.003 |
| | PARASOL | 0.019 | -0.061 | -0.085 | -0.032 | -0.031 | -0.048 | -0.028 | -0.326 | -0.039 | -0.085 | -0.033 | 0.004 | 0.077 |
| | SEVIRI | -0.165 | -0.038 | -0.226 | -0.201 | 0.026 | -0.004 | -0.126 | -0.218 | 0.046 | -0.034 | -0.005 | 0.019 | -0.105 |
| RMSE | Prior | 0.365 | 0.349 | 0.397 | 0.465 | 0.266 | 0.229 | 0.257 | 0.598 | 0.146 | 0.144 | 0.176 | 0.235 | 0.272 |
| | MODIS/Aqua | 0.264 | 0.172 | 0.306 | 0.28 | 0.143 | 0.109 | 0.191 | 0.607 | 0.087 | 0.142 | 0.112 | 0.129 | 0.264 |
| | MODIS/Terra | 0.258 | 0.167 | 0.307 | 0.28 | 0.145 | 0.108 | 0.193 | 0.604 | 0.086 | 0.14 | 0.109 | 0.155 | 0.258 |
| | MISR | 0.374 | 0.189 | 0.287 | 0.282 | 0.174 | 0.128 | 0.182 | 0.58 | 0.087 | 0.139 | 0.107 | 0.558 | 0.273 |
| | PARASOL | 0.381 | 0.21 | 0.295 | 0.286 | 0.158 | 0.144 | 0.226 | 0.566 | 0.094 | 0.152 | 0.112 | 0.294 | 0.478 |
| | SEVIRI | 0.273 | 0.156 | 0.329 | 0.288 | 0.155 | 0.113 | 0.205 | 0.518 | 0.11 | 0.132 | 0.1 | 0.188 | 0.261 |
| $\rho$ | Prior | 0.256 | 0.572 | 0.232 | 0.147 | 0.367 | 0.658 | 0.454 | 0.086 | 0.284 | 0.464 | 0.396 | 0.546 | 0.393 |
| | MODIS/Aqua | 0.465 | 0.67 | 0.384 | 0.307 | 0.379 | 0.716 | 0.589 | 0.439 | 0.465 | 0.463 | 0.665 | 0.532 | 0.452 |
| | MODIS/Terra | 0.468 | 0.685 | 0.357 | 0.28 | 0.393 | 0.718 | 0.569 | 0.432 | 0.458 | 0.447 | 0.669 | 0.496 | 0.472 |
| | MISR | 0.339 | 0.628 | 0.347 | 0.275 | 0.421 | 0.698 | 0.53 | 0.407 | 0.403 | 0.425 | 0.648 | 0.551 | 0.416 |
| | PARASOL | 0.274 | 0.635 | 0.374 | 0.308 | 0.403 | 0.676 | 0.537 | 0.434 | 0.406 | 0.431 | 0.647 | 0.382 | 0.205 |
| | SEVIRI | 0.486 | 0.676 | 0.286 | 0.348 | 0.415 | 0.681 | 0.495 | 0.406 | 0.461 | 0.43 | 0.683 | 0.274 | 0.484 |

**Table A1.** Statistics of the analyses against AERONET 500 nm AOD for selected sites. The acronym m.a.s.l. stands for meters above sea level, RMSE for root mean square error and $\rho$ is the Pearson correlation coefficient.

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
