# Peer review of "Impact of the choice of the satellite aerosol optical depth product in a sub-regional dust emission inversion"

_Atmospheric Chemistry and Physics, 2016_

## Referee Comment (RC1) · Anonymous Referee #1 · 13 Feb 2017

GENERAL COMMENTS:

This paper uses a state-of-the-art data assimilation system to investigate the influence of used satellite input into a dust emission inversion scheme. Inversion is still a relatively young field and it is therefore important to further develop existing systems and to test sensitivities. I therefore welcome this contribution to ACP. Overall the work is of high scientific quality and I have no issues with the content. However, to make this work more accessible for readers interested in dust emission, but not expert in inversion techniques, the authors should make more effort to improve the presentation, particularly the explanation of the methods. Moreover, the English is not always of highest standards; particularly the number of grammar errors (e.g. simple subject-verb

disagreements) and punctuation errors is annoyingly large.

MAJOR COMMENTS:

1) Introduction: To my taste it contains too much technical detail. Some of this could be moved to Section 2.

2) Section 2: With 4.5 pages, this is quite long for a Methods section of a relatively short paper. It is quite technical and a little hard to read. It would be good if the authors could spend a little more time trying to streamline this section and make it as didactic as possible, in order to make it more accessible for readers not so familiar with inversion techniques. I would start out with something like a road map, such that the reader knows what to expect. Then I would describe the model, then the obs, then the observation operator and finally the actual data assimilation. The way it is now is not logical in my eyes. Many readers will not know what the "control vector" is and introducing so early is a little hard to digest. Also the beginning of section 2.4 is hard to understand and the numbers given there all seem a little arbitrary.

MINOR COMMENTS:

1) Title: I would avoid an abbreviation in the title.

2) P1, L5: better have?

3) P1, L17-18: . . . combine model and observational information in the best possible way. Their application . . ..

4) P1, L18-19: In recent years, . . . AOD has also been . . .

5) P2, top: Add reference for Fe and P fertilisation!

6) P2, L5: Deposition into the ocean . . .

7) P2, L7: new paragraph after "quality." Then "Among other uncertainties . . ."

8) P2, L14: emission uncertainties

9) P2, L23: comma before respectively

10) P2, L24: However, MODIS products are not free of problems . . .

11) P2, L25: the MODIS aerosol product

12) P3, L24: referred to as SPLA

13) P3, L29: aerosol is

14) P3, L29: diameters less than . . . has diameters

15) P3, L31: aerosol tracer

16) P4, L4: were performed . . . ERA-Interim . . . as explained

17) P4, L12: tests . . . analysis to the grouping . . .

18) P4, L16: The same sub-regions as in EBCH16, defined depending on the emission category, are used.

19) P4, L18: map

20) P4, L19: have been defined: 15 over northern Africa, 3 over . . . the Middle East

21) P4, L26-29: Long and complicated sentence. Reword!

22) P4, L32: over the ocean

23) P5, L1: . . . instrument, as they . . .

24) P5, L15: . . . coverage, although . . . hence in the . . .

25) P5, L29: of the MISR algorithm

26) P6, L10: onto the model grid

27) P6, L18: . . . sample our region of interest only once per day.

28) P6, L19: . . . PARASOL), and so its . . .

29) P6, L23: standard deviation . . . is . . .

30) P6, L27: timescale gives

31) P6, L27: avoid repitition of words

32) P6, L33: was

33) Caption of F1: shown in the left column . . . in the right column. Please note the . . .

34) P7, L4: (EE), which

35) Table 1: What is Ck?

36) P8, L8: errors, which

37) P8, L11: error, assuming

38) P8, L15: These help to detect . . .

39) P8, L16: They assume that . . .

40) P8, L18: is better to draw . . .

41) P9, L2: more or less?? Reword!

42) P9, L3: retrieval dataset

43) P9, L5: where available, that is: . . .

44) P9, L14: refer back to methods section

45) P9, L16: super-coarse

46) Section 3.1: odd title

47) P9, L23: in the southern Red Sea

48) P9, L24: downwind of the . . . are hardly evident . . .

49) P9, L25: Atlantic is more extended than in the rest

50) P9, L26: Atlantic Ocean are found close to the . . .

51) P9, L26: yearly means for fine . . ..

52) P9, L27-28: remove brackets around lat-lon

53) P9, L31: To be able to roughly discriminate between the . . .

54) P9, L33: in Fig. 2. In this figure . . .

55) Fig. 2: caption too short, explain individual panels, ideally label them

56) P10, L6: relatively

57) P10, L7-8: in the south). However, total . . .Aqua in Fig. 2 is . . . counterpart in . . .

58) P11, L1: still hold

59) P11, L13: or in other words that the model . . .

60) P11, L16: counterparts

61) P11, L19: AODs (explained above) we think that . . . ; what makes you think so??

62) P11, L21: have total AOD available over land is PARASOL.

63) P11, L24: eastern Atlantic

64) P11, L29-30: plural of analysis is analyses! This part does not read very well.

65) Fig. 4: better "analysed AOD"? In the latter, we included the . . .

66) P13: I'm not sure I understand why it results in LARGE AOD values over land?!?

67) P14, L3: even though

68) P14, L 12 peaks in September

69) P14, L13: better "rule out" than "discard"

70) Fig. 5: Note that the three plots . . .

71) P15, L2: can be inferred to some extent from . . .

72) P15, L4: of the overall analysed

73) P16, L15: move "well" to end of sentence

74) P16, L17: capability to report

75) P16, L19: (Appendix A); the MISR . . .

76) P17, L4: some key model parameters . . .; which ones do you have in mind??

---

## Referee Comment (RC2) · Jerónimo Escribano et al. · 29 Mar 2017

The manuscript presents estimates of dust emission from Northern Africa and Arabian
Peninsula for the year 2006. Aerosol optical depth (AOD) retrievals from five different
satellite instruments are individually assimilated into a global model that includes a
simplified aerosol model. The individual assimilation allows to evaluate the spread of
the estimated dust emission due to the different AOD datasets. These are very interesting
and new results in the study, which should be published. Besides providing new
estimates for dust emission from Northern Africa and the Arabian Peninsula, which
are based on the assimilation, these results demonstrate that using only selected AOD
retrievals for estimating dust emission or model evaluation will likely lead to an underestimation
of the uncertainty in the results.

The structure of the manuscript needs improvement in some parts. The authors should also carefully revise with respect to the English language, especially the phrasing of some sentences.

Following points should particularly be taken into consideration before publication. Quotes from the manuscript are in italic:

1. **Abstract, lines 11–12:** *"We also show how the assimilation of a variety of AOD products can help to identify systematic errors in models".*

   It is not clear to me how the manuscript has shown such a thing as a guideline that can be generalized to other models. The authors make some short general statements in the conclusions of the manuscript about possible biases in the specific model that was applied by them, but that is not sufficient for such a general statement in the abstract.

   I recommend to remove the last sentence in the abstract.

   Alternatively, the authors could add a more systematic discussion of how the assimilation of the AOD retrievals can help identify model biases in general. This would further improve the paper.

2. **Page 2, lines 1–7**

   The relevant scientific references should be added to each of the points about the importance of dust aerosols.

3. **Page 2, line 27 to page 3, line 3**

   The scientific references for each of the listed instruments should be added.

4. **Page 4, lines 27–28:** *"...the use of efficient algorithms to ensure semi positiveness of some matrices involved in the inversion ..."*

For the purpose of reproducibility, it should be specified what algorithms were used in the current study to ensure this, instead of making a general statement only.

5. **Page 5, lines 11–12:** *Using this coefficient we derive the 550-nm AOD from these retrievals, for total and fine mode over ocean and fine mode over land.*

Even though it may appear trivial to the experts, the formula for deriving the 550-nm AOD should be presented here.

6. **Page 5, lines 28–29:** *"(i) we calculated the contribution of each aerosol model to the total AOD, using the reported fitting parameters and considering the 8 basic aerosol models of MISR algorithm;"*

This statement is not clear. Were only eight basic aerosol models out of the 74 aerosol mixture models considered and their contributions calculated? In any case, the sentence should be rephrased to clarify what was done.

7. **Page 5, lines 31–33:** *"In practice, our approximation of the AOD reprojected on the three modes of the SPLA model is accurate with a relative error of (maximum) 5% of the total AOD for the 5% less accurate recomputed retrievals"*

How was this relative error estimate derived? The information about the methodology how this relative error was obtained should be added to the manuscript.

8. **Page 7, lines 3–4:** *"The standard deviation of the observational errors have to be prescribed to the data assimilation system."*

This sounds more like an introductory statement to the discussed aspect and seems to be out of place in the structure here. It rather should be moved to the beginning of the paragraph.

9. **Page 8, lines 17–18:** *"..., so we decided not to inflate the covariance matrices."*

[Figure]

This has already been stated at the beginning of the paragraph. The repetition here is redundant, and it can be removed from the text.

10. **Page 8, lines 18–19:** *"Additionally, a common configuration for all the inversions is fairer to draw consistent conclusions across the five observational datasets."*

    This statement is a little bit difficult to understand. What does "fair" mean in this context here? Are the conclusions the ones that are consistent? Or does choosing a common configuration ensure a consistent approach for all the inversions to draw conclusions across the five observational datasets?

11. **Page 9, lines 3–14**

    This whole part is an introduction in the five satellite instruments that have been used for the assimilation. This part is presented after details of the treatment of the data from the instruments have already been discussed. It should be moved to the beginning of the section on the observations, before the details are discussed.

12. **Section "3. Results", Figures 1 and 2**

    Figures 1 and 2 present very interesting information about the differences between the AOD retrievals from the various satellites. One part of this information are the differences between the retrievals with respect to the relative fraction of the AOD that is coming from the fine mode relative to the total. However, this is difficult to evaluate from Figure 1 or 2, especially due to the different scales that are used for the fine mode AOD and the total AOD. I suggest to add a figure that displays the geographical distribution of the relative fractions of the fine mode AOD compared to the total AOD for the instruments for which it is available.

13. **Subsection "3.4 Mineral dust flux"**

    One result that is puzzling to me is the decrease in the mineral dust flux simulated with the model after assimilation, in the case of almost all satellite products

(except for PARASOL), even though the prior AOD in the model is on average lower than the AOD from the observations. This appears to be counterintuitive. If the model AOD increased after assimilation of the observations I would expect that this increase comes with a higher dust load and higher dust emission.

How do the authors explain this? This should be discussed in the manuscript.

**Language and typos:**

1. **Page 4, line 26:** Replace *"The later is mainly..."* with "The latter is mainly ...".

2. **Page 5, line 30:** Replace *"independent from"* with "independent of".

3. **Page 6, line 32:** Replace *"... difference with EBCH16 ..."* with "... difference to EBCH16 ...".

4. **Page 6, line 33:** Replace *"... the standard deviation of the observational errors were set to ..."* with "... the standard deviation of the observational errors was set to ..."

5. **Page 9, line 21:** Replace *"for year 2006"* with "for the year 2006".

6. **Page 9, lines 21–22:** *"Several characteristics can be identified in these yearly averages of AOD and they will impact the assimilation analysis."*

   I propose a rephrasing of the statement as follows: "Several characteristics that will impact the assimilation analysis can be identified in the yearly averages of the AOD."

---

## Author Comment (AC1)

**Response to Referee 1**

We would like to thank the referee for her/his helpful comments and remarks. We expect that the revised version will address all comments.

Motivated by the comment number 7 by referee 2, in this revised version we have revisited our post-processing algorithm of the MISR level 2 data. We no longer assume that the extinction efficiency is independent from the size of the aerosol and instead we compute the extinction efficiencies using the refractive index reported in the MISR products and a well-established Mie code. This improves the quality of the fine mode AOD derived from the MISR observations, but it decreases the fine mode AOD by approximately 15 %. The total AOD remains unchanged. We have recomputed the MISR analysis with this new dataset and we have included these new estimates in the revised version of the manuscript. The results for the MISR analysis only change marginally and the conclusions of the study remain the same.

We reproduce comments from the referee in "script" font followed by our answer. A document listing the revisions to the manuscript is also provided.

GENERAL COMMENTS:

This paper uses a state-of-the-art data assimilation system to investigate the influence of used satellite input into a dust emission inversion scheme. Inversion is still a relatively young field and it is therefore important to further develop existing systems and to test sensitivities. I therefore welcome this contribution to ACP. Overall the work is of high scientific quality and I have no issues with the content. However, to make this work more accessible for readers interested in dust emission, but not expert in inversion techniques, the authors should make more effort to improve the presentation, particularly the explanation of the methods. Moreover, the English is not always of highest standards; particularly the number of grammar errors (e.g. simple subject-verb disagreements) and punctuation errors is annoyingly large.

We thank the referee for all his/her comments and for the English corrections. We have included, at the beginning of Section 2, an overview of the data assimilation system.

MAJOR COMMENTS: 1) Introduction: To my taste it contains too much technical detail. Some of this could be moved to Section 2.

We appreciate the referee's comment, however we argue that the introduction does not contain too many technical details. The technical appearance is due to the relatively long list of satellites and instruments (and their acronyms) used to estimate AOD. Also according to the referee, Section 2 is already quite long.

2) Section 2: With 4.5 pages, this is quite long for a Methods section of a relatively short paper. It is quite technical and a little hard to read. It would be good if the authors could spend a little more time trying to streamline this section and make it as didactic as possible, in order to make it more accessible for readers not so familiar with inversion techniques. I would start out with something like a road map, such that the reader knows what to expect. Then I would describe the model, then the obs, then the observation operator and finally the actual data assimilation. The way it is now is not logical in my eyes. Many readers will not know what the "control vector" is and

introducing so early is a little hard to digest.  Also the beginning of section 2.4 is
hard to understand and the numbers given there all seem a little arbitrary.

We have included a new paragraph at the beginning of Section 2, which provides a roadmap to the four
subsections of Section 2. The paragraph reads:

"Mineral dust emissions are estimated using the source inversion system described in this Section. For-
mally, the combination of the a priori information, the AOD observations and the modelling system is
done through the minimization of the following cost function:

$$J(\mathbf{x}) = \frac{1}{2}(\mathbf{x} - \mathbf{x}^b)^T \mathbf{B}^{-1}(\mathbf{x} - \mathbf{x}^b) + \frac{1}{2}(\mathbf{y} - H(\mathbf{x}))^T \mathbf{R}^{-1}(\mathbf{y} - H(\mathbf{x})) \quad , \tag{1}$$

where the variable $\mathbf{x}$ is called the control vector and is related to the aerosol emissions (Sect. 2.2); $\mathbf{x}^b$
is the prior control vector, $\mathbf{y}$ are the assimilated observations (Sect. 2.3); $H$ is the observation operator
(Sect. 2.1); $\mathbf{B}$ is the covariance matrix of the background errors (Sect. 2.4); and $\mathbf{R}$ is the covariance
matrix of the observation errors (Sect. 2.4).

The solution of the minimization problem is called the analysis (denoted by $\mathbf{x}^a$). In this work the
*analysis AOD* is the observation operator evaluated for the analysis, that is, $H(\mathbf{x}^a)$. The components of
the inversion system (the elements of Eq. (1)) and the configuration of the data assimilation system are
now described."

Regarding the numbers given at the beginning of Sect. 2.4, we have included the following:

"The covariance matrix of the background errors ($\mathbf{B}$) and the covariance matrix of the observational
errors ($\mathbf{R}$) have to be prescribed in the data assimilation system. The $\mathbf{B}$ matrix is defined similarly to
EBCH16; the diagonal terms of the $\mathbf{B}$ matrix are defined using the error estimates presented in the work
of Huneeus et al. (2013). These are mostly based on the range of emissions found in the literature, except
for anthropogenic and fossil fuel emissions, which are based on the estimates of uncertainties found in
the literature. The standard deviation of the control vector errors (i.e., the square root of the diagonal
terms of $\mathbf{B}$) is 1.3 for biomass ..."

MINOR COMMENTS: 1) Title:  I would avoid an abbreviation in the title.

Done.

2) P1, L5:  better have?

Done.

3) P1, L17-18:  ...  combine model and observational information in the best possible
way.  Their application ...  .

Done.

4) P1, L18-19:  In recent years, ...  AOD has also been ...

Done.

5) P2, top:  Add reference for Fe and P fertilisation!

Done.

6) P2, L5:  Deposition into the ocean ...

Done.

7) P2, L7:  new paragraph after "quality."  Then "Among other uncertainties ..."

Done

8) P2, L14:  emission uncertainties

Done.

9) P2, L23:  comma before respectively

Done.

10) P2, L24:  However, MODIS products are not free of problems ...

Done.

11) P2, L25:  the MODIS aerosol product

We have changed the phrase to: "... the MODIS-DB aerosol product ... ".

12) P3, L24:  referred to as SPLA

Done.

13) P3, L29:  aerosol is

Done.

14) P3, L29:  diameters less than ...  has diameters

Done.

15) P3, L31:  aerosol tracer

Done.

16) P4, L4:  were performed ...  ERA-Interim ...  as explained

Done.

17) P4, L12:  tests ...  analysis to the grouping ...

Done.

18) P4, L16:  The same sub-regions as in EBCH16, defined depending on the emission category, are used.

Done.

19) P4, L18:  map

Done.

20) P4, L19:  have been defined:  15 over northern Africa, 3 over ...  the Middle East

Done.

21) P4, L26-29:  Long and complicated sentence.  Reword!

We have reformulated the sentence as follows:

"This results in a control vector of 4674 components (that is about 10 times larger than in EBCH16), with a **B** matrix of 4674 by 4674 elements (see Sect. 2.4). We have improved the data assimilation system presented in EBCH16 in order to deal with the larger control vector. To this effect we have carefully recoded some matrix multiplication and inversion routines, paying special attention to the computational memory management and minimizing numerical errors as much as possible. We have also applied the algorithm of Qi and Sun (2006) to ensure the semi-positiveness of some of the matrices involved in the inversion."

```
22) P4, L32:  over the ocean
```

Done.

```
23) P5, L1:  ...  instrument, as they ...
```

Done.

```
24) P5, L15:  ...  coverage, although ...  hence in the ...
```

Done.

```
25) P5, L29:  of the MISR algorithm
```

Done.

```
26) P6, L10:  onto the model grid
```

Done.

```
27) P6, L18:  ...  sample our region of interest only once per day.
```

Done.

```
28) P6, L19:  ...  PARASOL), and so its ...
```

Done.

```
29) P6, L23:  standard deviation ...  is ...
```

Done.

```
30) P6, L27:  timescale gives
```

Done.

```
31) P6, L27:  avoid repitition of words
```

We have replaced the sentence by:

"In comparison with EBCH16, this shortened timescale gives more freedom to the inversion system. Along with the three day sub-periods, this timescale allows the system to have more control over the emissions, with the aim of improving the representation of individual dust events in the analysis."

```
32) P6, L33:  was
```

Done.

```
33) Caption of F1:  shown in the left column ...  in the right column.  Please note the
...
```

Done.

```
34) P7, L4:  (EE), which
```

Done.

```
35) Table 1:  What is Ck?
```

$C_k$ is the error reported in the AERUS-GEO product. It is computed in the AERUS-GEO retrieval algorithm. We have added the following to the Table caption:

"Errors for the SEVIRI dataset $(C_k)$ are reported along with the AERUS-GEO AOD product and they are described in Carrer et al. ..."

```
36) P8, L8:  errors, which
```

Done.

```
37) P8, L11:  error, assuming
```

Done.

```
38) P8, L15:  These help to detect ...
```

Done.

```
39) P8, L16:  They assume that ...
```

Done.

```
40) P8, L18:  is better to draw ...
```

We have replaced this sentence with the following:

"Additionally, a common configuration for all the inversions ensures a consistent methodological approach to compare the five data assimilation experiments."

```
41) P9, L2:  more or less??  Reword!
```

We have deleted "more or less".

```
42) P9, L3:  retrieval dataset
```

Done.

```
43) P9, L5:  where available, that is:  ...
```

Done.

```
44) P9, L14:  refer back to methods section
```

Done.

```
45) P9, L16:  super-coarse
```

Done.

```
46) Section 3.1:  odd title
```

We have changed the title.

47) P9, L23:  in the southern Red Sea

Done.

48) P9, L24:  downwind of the ...  are hardly evident ...

Done.

49) P9, L25:  Atlantic is more extended than in the rest

Done.

50) P9, L26:  Atlantic Ocean are found close to the ...

Done.

51) P9, L26:  yearly means for fine ....

Done.

52) P9, L27-28:  remove brackets around lat-lon

Done.

53) P9, L31:  To be able to roughly discriminate between the ...

Done.

54) P9, L33:  in Fig.  2.  In this figure ...

Done.

55) Fig.  2:  caption too short, explain individual panels, ideally label them

We have expanded the caption in concordance with the caption of Fig. 1.

56) P10, L6:  relatively

Done.

57) P10, L7-8:  in the south).  However, total ... Aqua in Fig.  2 is ...  counterpart
in ...

Done.

58) P11, L1:  still hold

Done.

59) P11, L13:  or in other words that the model ...

Done.

60) P11, L16:  counterparts

Done.

61) P11, L19:  AODs (explained above) we think that ...  ; what makes you think so??

We have expanded our explanation as follows:

"We recall that the prior simulation is the same for all panels, and the difference in prior lies in the local time and gridboxes for which the model values are sampled. We have shown in Sect. 3.1 that, even for colocated retrievals, the geographical distribution of the AOD varies between the satellite products. We think that these differences contribute more to the differences between the histograms of Fig. 3 than the sampling differences. For example, the MODIS/Terra AOD of Fig. 1 is qualitatively similar to the MODIS/Terra AOD of Fig. 2, where only a subset of observations (which are coincident with MISR retrievals) is taken into account. On the contrary, it is easier to qualitatively observe the differences between the MISR and the MODIS/Terra panels of Fig. 2 (where both panels have the sample sampling)."

62) P11, L21: have total AOD available over land is PARASOL.

Done.

63) P11, L24: eastern Atlantic

Done.

64) P11, L29-30: plural of analysis is analyses! This part does not read very well.

We have corrected the paragraph accordingly.

65) Fig. 4: better "analysed AOD"? In the latter, we included the ...

We have rewritten the sentence:

"Simulated AOD at 550 nm for the prior and for the five analyses ..."

66) P13: I'm not sure I understand why it results in LARGE AOD values over land?!?

We have expanded the paragraph by the following:

"The SEVIRI analysis shows a larger transatlantic dust plume in MAM and JJA along with larger values of AOD over land. Observational uncertainties for SEVIRI are generally larger over land than over ocean. This allows the assimilation system to favour a better fit of the AOD over the ocean than over land. Over the transatlantic dust plume, the assimilated AOD is larger than the prior AOD. The analysis decreases this AOD difference by increasing the dust emissions in West Africa, and therefore the SEVIRI analysis shows larger AOD values over land."

67) P14, L3: even though

Done.

68) P14, L 12 peaks in September

Done.

69) P14, L13: better "rule out" than "discard"

Done.

70) Fig. 5: Note that the three plots ...

Done.

71) P15, L2: can be inferred to some extent from ...

Done.

```
72) P15, L4:  of the overall analysed
```

Done.

```
73) P16, L15:  move "well" to end of sentence
```

Done.

```
74) P16, L17:  capability to report
```

Done.

```
75) P16, L19:  (Appendix A); the MISR ...
```

Done.

```
76) P17, L4:  some key model parameters ...; which ones do you have in mind??
```

We have realised that the model parameter optimization could be hard to accomplish, mostly because of the difficulties in defining the **B** matrix properly. Instead, we have opened a different perspective that now reads:

"... Another approach which we leave for future work would be to estimate the net aerosol fluxes, that is, including variables related to the aerosol removal processes in the control vector. It would be interesting to explore this approach, since bias in the aerosol removal processes could introduce bias in the emissions if only the emissions are optimised; but the implementation of this data assimilation could be be difficult to accomplish, due to the increase in the degrees of freedom in an ill-posed data assimilation problem."

**References**

Huneeus, N., Boucher, O., and Chevallier, F. (2013). Atmospheric inversion of $SO_2$ and primary aerosol emissions for the year 2010. *Atmospheric Chemistry and Physics*, 13(13):6555–6573.

Qi, H. and Sun, D. (2006). A quadratically convergent newton method for computing the nearest correlation matrix. *SIAM Journal on Matrix Analysis and Applications*, 28(2):360–385.

---

## Author Comment (AC2)

**Response to Referee 2**

We would like to thank the referee for her/his helpful comments and remarks. We expect that the revised version will address all comments.

Motivated by the referee's comment number 7, in this revised version we have revisited our post-processing algorithm of the MISR level 2 data. We no longer assume that the extinction efficiency is independent from the size of the aerosol and instead we compute the extinction efficiencies using the refractive index reported in the MISR products using a well-established Mie code. This improves the quality of the fine mode AOD derived from the MISR observations, but it decreases the fine mode AOD by approximately 15 %. The total AOD remains unchanged. We have recomputed the MISR analysis with this new dataset and we have included these new estimates in the revised version of the manuscript. The results for the MISR analysis only change marginally and the conclusions of the study remain the same.

We reproduce comments from the referee in "script" font followed by our answer. A document listing the revisions to the manuscript is also provided.

The manuscript presents estimates of dust emission from Northern Africa and Arabian Peninsula for the year 2006. Aerosol optical depth (AOD) retrievals from five different satellite instruments are individually assimilated into a global model that includes a simplified aerosol model. The individual assimilation allows to evaluate the spread of the estimated dust emission due to the different AOD datasets. These are very interesting and new results in the study, which should be published. Besides providing new estimates for dust emission from Northern Africa and the Arabian Peninsula, which are based on the assimilation, these results demonstrate that using only selected AOD retrievals for estimating dust emission or model evaluation will likely lead to an underestimation of the uncertainty in the results.

The structure of the manuscript needs improvement in some parts. The authors should also carefully revise with respect to the English language, especially the phrasing of some sentences.

We have done our best to revise the English language of the manuscript. We have also implemented all language corrections requested by Referee 1.

Following points should particularly be taken into consideration before publication. Quotes from the manuscript are in italic:

1. Abstract, lines 11-12: "We also show how the assimilation of a variety of AOD products can help to identify systematic errors in models".

It is not clear to me how the manuscript has shown such a thing as a guideline that can be generalized to other models. The authors make some short general statements in the conclusions of the manuscript about possible biases in the specific model that was applied by them, but that is not sufficient for such a general statement in the abstract.

I recommend to remove the last sentence in the abstract.

Alternatively, the authors could add a more systematic discussion of how the assimilation of the AOD retrievals can help identify model biases in general. This would further improve the paper.

We agree with the referee that this statement is not enough substantiated so we have removed this sentence from the abstract. We have however left the corresponding discussion in the main text.

**2. Page 2, lines 1-7**

The relevant scientific references should be added to each of the points about the importance of dust aerosols.

We have added more references.

**3. Page 2, line 27 to page 3, line 3**

The scientific references for each of the listed instruments should be added.

We have added the scientific references for the AOD products listed in the paragraph (when available).

4. **Page 4**, **lines 27-28**: "...the use of efficient algorithms to ensure semi-positiveness of some matrices involved in the inversion ..."

For the purpose of reproducibility, it should be specified what algorithms were used in the current study to ensure this, instead of making a general statement only.

We have modified the paragraph and have included the appropriate reference. The new paragraph reads:

"We have improved the data assimilation system presented in EBCH16 in order to deal with the longer control vector. To this effect we have carefully recoded some matrix multiplication and inversion routines, paying special attention to the computational memory management and minimizing numerical errors as much as possible. We have also applied the algorithm of Qi and Sun (2006) to ensure the semi-positiveness of some of the matrices involved in the inversion."

5. Page 5, lines 11-12: Using this coefficient we derive the 550-nm AOD from these retrievals, for total and fine mode over ocean and fine mode over land.

Even though it may appear trivial to the experts, the formula for deriving the 550-nm AOD should be presented here.

**We have added the formula:**

"...over ocean and fine mode over land. That is, we interpolate the AOD using the following relation:

$$\tau_{550} = \tau_{865} \left(\frac{550}{865}\right)^{-\alpha} \tag{1}$$

where  $\tau_{550}$  is the AOD at 550 nm,  $\tau_{865}$  the AOD at 865 nm and  $\alpha$  is the Angström coefficient between 670 and 865 nm."

6. **Page 5, lines 28-29:** "(*i*) we calculated the contribution of each aerosol model to the total AOD, using the reported fitting parameters and considering the 8 basic aerosol models of MISR algorithm;"

This statement is not clear. Were only eight basic aerosol models out of the 74 aerosol mixture models considered and their contributions calculated? In any case, the sentence should be rephrased to clarify what was done.

Each of the 74 aerosol models is a mixture (or weighted sum) of 3 basic aerosol models. The list of mixture and basic aerosol models can be found in Kahn and Gaitley (2015). To clarify this point, we have rephrased the sentence and added this information in the previous sentence:

"... radiances for each observed pixel, and the quality of the fit is estimated using a chi-square criteria (Kahn et al., 2005). Each aerosol mixture model is composed by the weighted sum of (at most) three

basic aerosol models. The optical properties, the two parameters of the log-normal size distribution and the relative contributions of each basic aerosol model to the mixture aerosol models are reported in the Level 2 of the MISR products along with the fitting parameters computed in the AOD retrieval. With this information and with the reported Level 2 AOD, we have calculated an estimate of the MISR 555 nm AOD with the same diameter cut-off than the SPLA model, i.e., for fine (less than 1 µm of diameter), coarse (between 1 and 6 µm of diameter) and super-coarse (larger than 6 µm of diameter) aerosols. Briefly, the post-processing of the MISR AOD consists of the following steps: (i) we calculated the contribution of each basic aerosol model to the total AOD for each observed pixel; (ii) assuming that both, the reported refractive index for each model is independent of the size distribution, and the aerosol particles are spherical; we estimated the contribution of each bin (as the SPLA definitions) to the total AOD. In this work we only used the recomputed fine mode and total 555 nm MISR AOD."

7. **Page 5, lines 31-33:** "In practice, our approximation of the AOD reprojected on the three modes of the SPLA model is accurate with a relative error of (maximum) 5% of the total AOD for the 5% less accurate recomputed retrievals"

How was this relative error estimate derived? The information about the methodology how this relative error was obtained should be added to the manuscript.

We thank the referee for this question. We have taken the opportunity to recompute the MISR AOD with a better reprojection method, and we have updated the manuscript accordingly.

The updated method computes the optical properties of the aerosol populations using a Mie code, and we think that this is a better option than the one used in the previous version of the manuscript. To assert the accuracy of the reprojection method, the only possible reference dataset is the reported MISR small, medium and large AODs. These AODs are only published in their Level 3 product, while the reprojection process has to occur at the Level 2 stage.

In contrast to the total AOD, we did not find any documentation for the computation of the Level 3 of these AODs (by bin of size) in the MISR product. Thus, we assume that MISR computes the Level 3 small, medium and large AOD in the same way as the total AOD, that is, with the *mean* estimator of the Level 2 products.

Even if we cannot directly evaluate our reprojected AOD, a table is available ("Mixture Fractional Spectral Optical Depth Per Classification" from the Level 2 products) which relates each mixture model with their contribution (in AOD) to the estimate of small, medium and large AODs. These parameters are written in terms of relative contributions to the total AOD, that is, for each mixture model, the sum of the 3 parameters is unity. We have compared this table with an equivalent table computed through our Mie code. We have found that the differences in the values of these tables are small (less than 0.0035 in the table). These comparison indicates that is it possible to recompute the Level 2 AODs with an acceptable accuracy.

As we cannot completely simulate the MISR Level 3 product (because of the lack of documentation explained above), we do not expect that our approximation exactly matches with the reported Level 3 AODs. In fact, the RMSE between the AODs (for all of them, small, medium and large AODs) is close to 0.02, and the bias is not significant. The total AOD is not affected by this error.

In consequence, we have removed the sentence from the manuscript, as it refers to the accuracy in the recomputation of the total AOD in the Level 2 products from the previous version of the manuscript.

8. **Page 7, lines 3-4:** "The standard deviation of the observational errors have to be prescribed to the data assimilation system."

This sounds more like an introductory statement to the discussed aspect and seems to be out of place in the structure here. It rather should be moved to the beginning of the paragraph.

The referee is correct and we have added this information at the beginning of the section.

9. Page 8, lines 17-18: "..., so we decided not to inflate the covariance matrices."

This has already been stated at the beginning of the paragraph. The repetition here is redundant, and it can be removed from the text.

We have followed the referee's recommendation.

10. **Page 8, lines 18-19:** "Additionally, a common configuration for all the inversions is fairer to draw consistent conclusions across the five observational datasets."

This statement is a little bit difficult to understand. What does "fair" mean in this context here? Are the conclusions the ones that are consistent? Or does choosing a common configuration ensure a consistent approach for all the inversions to draw conclusions across the five observational datasets?

We agree with the referee's comment. We have clarified this point in the modified manuscript:

"Additionally, a common configuration for all the inversions ensures a consistent methodological approach to compare the five data assimilation experiments."

**11. Page 9, lines 3-14**

This whole part is an introduction in the five satellite instruments that have been used for the assimilation. This part is presented after details of the treatment of the data from the instruments have already been discussed. It should be moved to the beginning of the section on the observations, before the details are discussed.

We have moved this part to the beginning of the section.

**12. Section "3. Results", Figures 1 and 2**

Figures 1 and 2 present very interesting information about the differences between the AOD retrievals from the various satellites. One part of this information are the differences between the retrievals with respect to the relative fraction of the AOD that is coming from the fine mode relative to the total. However, this is difficult to evaluate from Figure 1 or 2, especially due to the different scales that are used for the fine mode AOD and the total AOD. I suggest to add a figure that displays the geographical distribution of the relative fractions of the fine mode AOD compared to the total AOD for the instruments for which it is available.

We have followed the referee's recommendation and we have added a third column in Figures 1 and 2. We are aware that the different color scales make the comparison harder, but we have included a note in the caption of the figure, indicating that the color scale of the fine AOD is exactly half of that of the total AOD, making an easier comparison. The caption now reads :

"Averages for the year 2006 of the satellite-derived AOD products used in this study. The AOD products are all regridded to a regular latitude-longitude grid of 0.5° resolution for MISR and SEVIRI and 1° for

MODIS and PARASOL. The total AOD is shown in the left column, the fine mode AOD (when available) in the middle column, and the ratio between the average fine mode AOD and the average total AOD is shown in the right column. Please note the 2:1 ratio of the color scales between the left (total AOD) and middle (fine model AOD) columns and the (somewhat) different wavelengths of the reported AODs."

**13. Subsection "3.4 Mineral dust flux"**

One result that is puzzling to me is the decrease in the mineral dust flux simulated with the model after assimilation, in the case of almost all satellite products (except for PARASOL), even though the prior AOD in the model is on average lower than the AOD from the observations. This appears to be counter-intuitive. If the model AOD increased after assimilation of the observations I would expect that this increase comes with a higher dust load and higher dust emission.

How do the authors explain this? This should be discussed in the manuscript.

The referee is right and this behaviour of the analysis seems indeed counter-intuitive. We have to clarify why, on average, the analysis AOD is lower than the prior AOD (Figure 4 and more quantitatively in Table A1), which is consistent with the decrease of the dust emissions after the assimilation. Even though the observational AOD is larger than the prior, the data assimilation system attempts to decrease preferentially the extremes of the departure distribution. This decrease of the departures is more effective on the left side of the histograms. The preferential decrease for the extremes of the distribution is due to the formulation of the cost function, where the distance to be minimized is related to the square of the departures and thus this is the preferred behaviour.

Additionally, the construction of the control vector does not allow creating emissions if the dust production module does not produce them in the prior. We have done a qualitative comparison between the assimilated, prior and analysis AOD at the daily resolution. This comparison suggests that the larger prior departures of AOD, that is, when the observations are larger than the prior AOD, are mostly due to the dust produced in individual dust events which are not simulated by the prior. With the current configuration of the data assimilation system, these departures cannot be decreased in the analysis. In summary, the system "easily" decreases the largest model overestimations of AOD, but it has a hard job to increase the largest model underestimations of AOD. This is also reflected in the decrease in the mean simulated AOD and the increase in the bias (in comparison with AERONET AOD).

We have included in Fig. 1 of this document, a frequency plot for one of the experiments, which illustrates the decrease of the left tail of the departure distribution. It is possible to observe, by comparing the first and second columns, that the large departures in the upper-left region of the Obs. vs Prior (that is, large prior and small observational AOD) panels are decreased in the Obs. vs Analysis panels, while the large departures in the lower-right region (small prior and large observational AOD) are not decreased.

The manuscript has been changed accordingly. We have included the following paragraph in Section 3.2:

"A common feature is observed in all the analyses of Fig. 3, which is the preferential decrease of the left tail of the departure distributions after the assimilation. In other words, the data assimilation system is more efficient (in terms of minimizing the cost function) in decreasing larger values of model AOD than in increasing small values of model AOD. The reason for this preference is linked to the constraints imposed by the dust production model and also to the definition of the control vector. The dust production module emits dust only if some conditions are met, for example, only when there is no vegetation, the wind speed is above a threshold value (depending on the soil texture), etc. These conditions are parameterised in

the model, so they depend on the model performance, but it is important to note that these conditions are based on the physical mechanisms of the natural emissions of dust. The control vector is, in practice, a multiplicative factor for the aerosol emissions. If the dust production model has no positive emission flux, the analysis cannot increase these emissions. On the contrary, if the dust emission flux is too large, the analysis can decrease the emissions. In consequence, we think that the preferential decrease of the left tail of the departure distributions is due to deficiencies of the prior in simulating some dust emissions events."

We have included the following comment at the end of Section 3.2:

"... We would like to stress that, even though the mode of the departures is closer to zero in the analyses, the average of the departures is not necessarily closer to zero. For MODIS/Aqua, MODIS/Terra and MISR, the average of the departures for the *all* curve of Fig. 3 is larger in the analyses than in the prior. This means that for these experiments (as the average of the prior departures is positive), the average AOD in the analyses is smaller than the prior AOD. This is exemplified in the comparison with AERONET, in the Appendix A, and will be related with the overall decrease of analysed emissions in Sect. 3.4."

And the following comment in Section 3.4:

"...for the super-coarse dust emission panel. The decrease of emissions of the analyses with respect to the prior is consistent with the results discussed in Sect. 3.2, where the average AOD is smaller in the analysis than in the prior, for the simulated AOD coincident with the assimilated observations for the MODIS and MISR experiments."

Language and typos:

1. Page 4, line 26: Replace "The later is mainly..." with "The latter is mainly ...".

We have replaced this paragraph with the following (already written in comment number 4):

"We have improved the data assimilation system presented in EBCH16 in order to deal with the larger control vector. To this effect we have carefully recoded some matrix multiplication and inversion routines, paying special attention to the computational memory management and minimizing numerical errors as much as possible. We have also applied the algorithm of Qi and Sun (2006) to ensure the semi-positiveness of some of the matrices involved in the inversion."

2. Page 5, line 30: Replace "independent from" with "independent of".

Done.

3. Page 6, line 32: Replace "... difference with EBCH16 ..." with "... difference to EBCH16 ...".

Done.

4. **Page 6, line 33:** Replace "... the standard deviation of the observational errors were set to ..." with "... the standard deviation of the observational errors was set to ..."

Done.

5. Page 9, line 21: Replace "for year 2006" with "for the year 2006".

Done.

6. **Page 9, lines 21-22:** "Several characteristics can be identified in these yearly averages of AOD and they will impact the assimilation analysis."

I propose a rephrasing of the statement as follows: "Several characteristics that will impact the assimilation analysis can be identified in the yearly averages of the AOD."

We thank the referee for the suggestion. The sentence has been modified accordingly.